# 4M-21: An Any-to-Any Vision Model for Tens of Tasks and Modalities

**Roman Bachmann**[1][†][*]    **Oğuzhan Fatih Kar**[1][*]    **David Mizrahi**[2][†][*]    **Ali Garjani**[1]

**Mingfei Gao**[2]    **David Griffiths**[2]    **Jiaming Hu**[2]    **Afshin Dehghan**[2]    **Amir Zamir**[1]

[1]Swiss Federal Institute of Technology Lausanne (EPFL)    [2]Apple

https://4m.epfl.ch

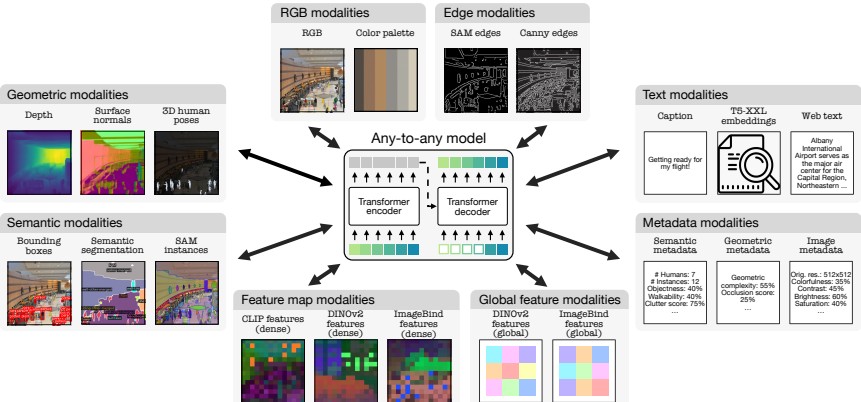

Figure 1: We demonstrate training a single model on tens of highly diverse modalities *without a loss in performance* compared to specialized single/few task models. The modalities are mapped to discrete tokens using modality-specific tokenizers. The model can generate *any* of the modalities from *any subset* of them.

## Abstract

Current multimodal and multitask foundation models, like 4M [65] or Uni-fiedIO [61, 60], show promising results. However, their out-of-the-box abilities to accept diverse inputs and perform diverse tasks are limited by the (usually small) number of modalities and tasks they are trained on. In this paper, we develop a single any-to-any model trained on tens of highly diverse modalities and by performing co-training on large-scale multimodal datasets and text corpora. This includes training on images and text along with several semantic and geometric modalities, feature maps from recent state of the art models like DINOv2 and ImageBind, pseudo labels of specialist models like SAM and 4DHumans, and a range of new modalities that allow for novel ways to interact with the model and steer the generation, for example, image metadata or color palettes. A crucial step in this process is performing discrete tokenization on various modalities, whether they are image-like, neural network feature maps, vectors, structured data like instance segmentation or human poses, or data that can be represented as text.

Through this, we show the possibility of training one model to solve at least 3x more tasks/modalities than existing models and doing so *without a loss in performance*. In addition, this enables more fine-grained and controllable multimodal generation capabilities and allows studying the distillation of models trained on diverse data and objectives into one unified model. We scale the training to a three billion parameter and different datasets. The multimodal models and training code are open sourced at https://4m.epfl.ch.

---
[*]Equal contribution & corresponding authors. Randomized order.
[†]Work partially done while at EPFL and Apple.

38th Conference on Neural Information Processing Systems (NeurIPS 2024).

# 1 Introduction

Having *a single neural network* to handle a wide and varied range of tasks and modalities has been a longstanding goal. Such a model, especially when capable of any-to-any predictions, brings notable advantages, such as test-time computational efficiency, model size, and enabling modality fusion.

However, multitask learning has commonly faced significant challenges. For example, the training often suffers from negative transfer, leads to reduction in performance compared to single-task models, and typically requires careful strategies for balancing losses or gradients [48, 99, 86, 100, 33]. Moreover, training a single network on tasks and modalities that vary greatly in terms of dimensionality, data type, and value ranges presents additional complexities[†]. Recent notable efforts in the space of multimodal and multitask training, such as Pix2Seq [17, 18], OFA [93], 4M [65], or Unified-IO [61, 60] have made significant strides in unifying the representation space for conceptually different inputs and targets. A large part of their success can be attributed to transforming different modalities into a common representation, namely sequences of discrete tokens, and training relatively standard Transformer architectures on them. While these works show promising results, they are typically trained on a small set of modalities. This raises the question if increasing the set of tasks/modalities the models can solve will lead to a degradation of performance.

We build upon the multimodal masking pre-training scheme [65] and increase its capabilities by training on tens of highly diverse modalities. Concretely, we add SAM segments [49], 3D human poses and shapes from 4DHumans [37], canny edges extracted from RGB and SAM instances, color palettes, multiple types of image, semantic and geometric metadata, as well as T5-XXL [71] text embeddings, in addition to 7 more common modalities. On top of that, we include dense feature maps of the recent state of the art models DINOv2 [68] and ImageBind [35], as well as their global embedding vectors to enable multimodal retrieval abilities. Please see fig. 1 for an overview.

We are able to train a single unified model on diverse modalities by encoding them with modality-specific discrete tokenizers (see fig. 3). For image-like modalities, e.g. RGB or edges, we train ViT-based [25] VQ-VAE [66] tokenizers to map the inputs into a small grid of discrete tokens. For modalities like 3D human poses or image embeddings, we train MLP-based discrete VAEs to compress them into a small set of discrete tokens. All other modalities that can be mapped to a text representation, such as captions or metadata, are encoded using a WordPiece tokenizer [24].

The resulting model demonstrates the possibility of training a single model on a large number of diverse modalities/tasks without any degradation in performance and significantly expands the out-of-the-box capabilities compared to existing models. Adding all these modalities enables new potential for multimodal interaction, such as retrieval from and across multiple modalities, or highly steerable generation of any of the training modalities, all by a single model.

In short, we expand the capabilities of existing models across several key axes:

- **Modalities**: Increase from 7 in the existing best any-to-any models [65] to 21 diverse modalities, enabling new capabilities like cross-modal retrieval, controllable generation, and strong out-of-the-box performance. This is one of the first times in the vision community that a single model can solve **tens of diverse tasks in an any-to-any manner** (see fig. 2), without sacrificing performance and especially do so without any of the conventional multitask learning difficulties [77, 48, 99, 86, 100, 33].

- **Diversity**: Add support for more structured data, such as human poses, SAM instances, metadata, and color palettes for controllable generation.

- **Tokenization**: Investigate discrete tokenization of diverse modalities such as global image embeddings, human poses, and semantic instances using modality-specific approaches.

- **Scale**: Scale the model size to 3B parameters and dataset to 0.5B samples using [12].

- **Co-Training**: Demonstrate co-training on vision and language modeling simultaneously.

---

[†]Modality vs task: "Modalities" usually denote the *inputs* to a model (e.g. sensory signals), and "tasks" usually denote the *outputs* (e.g. semantics). The adopted architecture in multimodal masked modeling enables a symmetric input-output structure, thus modalities and tasks are used interchangeably in this paper.

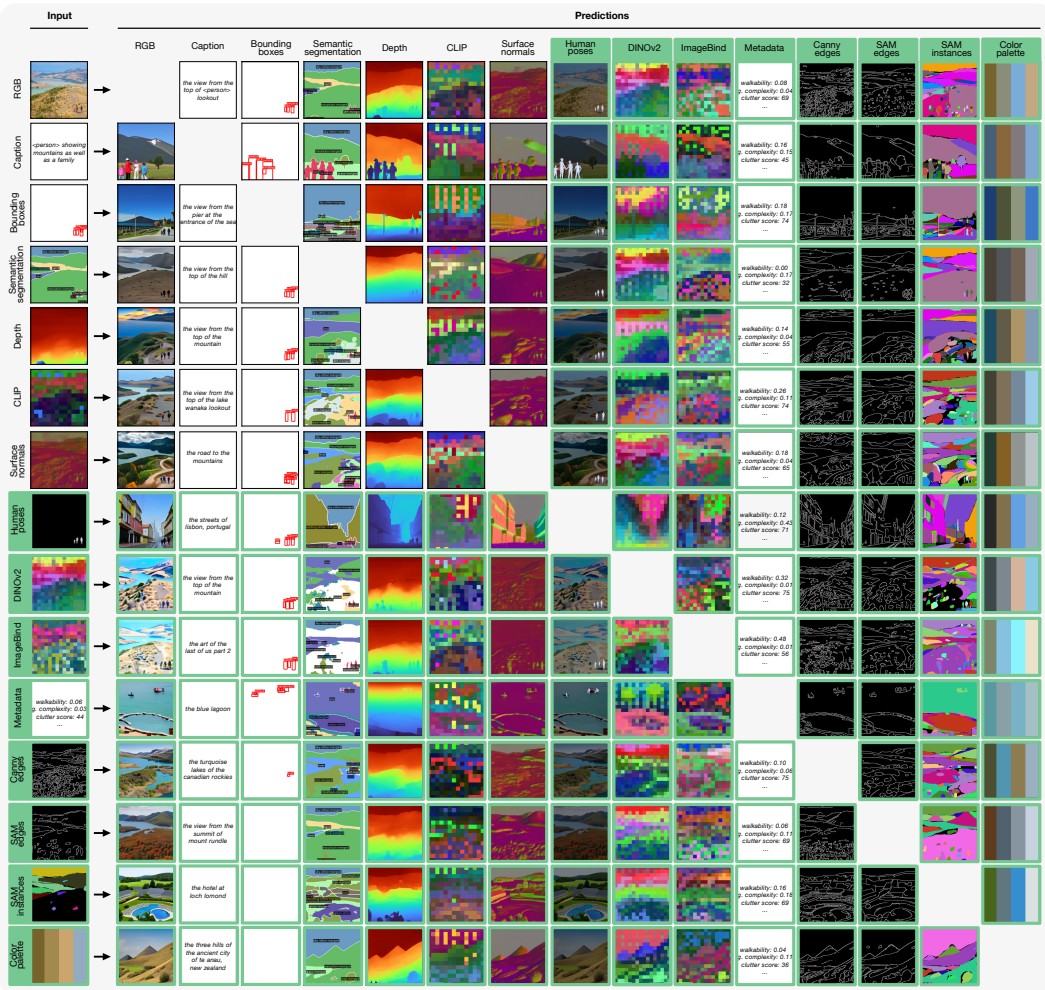

Figure 2: **One-to-all generation.** 4M-21 can generate all modalities from any given input modality and can benefit from chained generation [65]. Notice the *high consistency* among the predictions of all modalities for one input. Each row starts from a different modality coming from the same scene. Highlighted in green are new input/output pairs that 4M [65] cannot predict nor accept as input. Note that, while this figure shows predictions from a single input, 4M-21 can generate any modality from *any subset of all modalities*.

## 2 Method

We adopt the 4M pre-training scheme [65] as it has been shown to be a versatile approach that can be efficiently scaled to a diverse set of modalities. We keep the architecture and the multimodal masked training objective the same, but expand upon the model and dataset size, the types and number of modalities with which we train the model, and train jointly on multiple datasets. All modalities are first transformed into sequences of discrete tokens using modality-specific tokenizers (See fig. 3). During training, random subsets of these tokens are selected from all modalities as inputs and targets, and the objective is to predict one subset from the other. We rely on pseudo labeling to create a large pre-training dataset with multiple aligned modalities. See appendix I.1 for a discussion on different architecture choices.

### 2.1 Modalities

We train on a large and diverse set of modalities that we group into the following categories: RGB, geometric, semantic, edges, feature maps, metadata, and text modalities. Below we provide a summary of them (See fig. 1 and appendices D and E for details, and fig. 2 for generation examples).

**RGB:** We include both tokenized and pixel versions of RGB images to facilitate transfer learning. In particular, discrete tokens enable iterative sampling, making them useful for generative tasks [96, 15].

On the other hand, using RGB pixels as input is more suitable for visual perception tasks. By avoiding the discrete bottleneck, there is no information loss during the tokenization step, and the projection layer can be more lightweight. Given these tradeoffs, we follow 4M by training on both and treating them as separate modalities, with RGB pixels as an input-only modality. We also extracted *color palettes* from RGB images using PyPalette [2], at varying number of colors. This enables us to perform conditional generation using desired colors for better artistic control.

**Geometric modalities:** These contain *surface normals*, *depth*, and *3D human poses & shape* which provide important information about the scene geometry. For the first two, we used Omnidata models from [27, 46] for pseudo labeling due to their strong generalization performance. For 3D human poses and shape, we leverage a recent state-of-the-art model, 4D-Humans [37].

**Semantic modalities:** We include *semantic segmentation* and *bounding boxes* to capture the scene semantics and leverage Mask2Former [20] and ViTDet [56] models for pseudo labeling. Next to these, we also incorporated pseudo labels extracted from Segment Anything Model [49] (SAM) as *SAM instances* for its strong object representation.

**Edges:** As recent generative methods such as ControlNet [104] showed, edges carry important information about the scene layout and semantics that are also useful for conditioning, abstraction, and sketching. We consider two types of edges, specifically *Canny edges* and *SAM edges*. The former is extracted from the RGB images with OpenCV [1]. As Canny edges may contain low-level information, e.g. shading edges, we also include edges extracted from SAM instances to get a more semantic boundary map. We tokenize Canny and SAM edges with a shared tokenizer.

**Feature maps:** We extract embeddings from *CLIP* [70], *DINOv2* [68] and *ImageBind* [35] as they demonstrated strong transfer learning and retrieval capabilities. Previously, tokenized CLIP features were shown to be an effective target for masked image modelling [94, 65] that enables distilling a useful semantic representation of the scene. We follow a similar approach and tokenize the feature maps from pre-trained CLIP-B16, DINOv2-B14 and ImageBind-H14 models. We also included the *global embeddings* of DINOv2 and ImageBind models and tokenized them separately.

**Metadata:** We extract several useful pieces of information from the RGB images and other modalities, that can be categorized into *semantic metadata*, *geometric metadata*, and *image processing metadata*. For this, we use functionalities from Pillow [3] OpenCV [1], and Omnidata [27].

The following semantic metadata are extracted from bounding boxes, poses, and segmentation maps:

- *Crowdedness score*: number of humans (extracted from 4DHumans instances)
- *SAM clutter score*: number of SAM instances
- *COCO clutter score*: number of COCO [57] instances
- *COCO instance diversity*: number of unique COCO instance classes
- *Objectness score*: % of pixels that belong to countable COCO semantic classes
- *Walkability score*: % of pixels belonging to walkable COCO semantic classes such as 'road'
- *Semantic diversity*: number of unique COCO semantic classes
- *Caption length*: length of the caption in characters, words, and sentences

These are aimed to capture the semantic regularities of the scene at a more holistic level as opposed to pixel-based representations.

Similarly, geometric metadata captures the scene geometry more globally. They are extracted from surface normals and depth maps:

- *Geometric complexity*: angular variance of surface normals
- *Occlusion score*: % of occlusion edges over a fixed threshold

Finally, image processing metadata contains several aspects of images such as *original image height and width* before cropping, which can be used as conditioning to generate higher quality images [69], *brightness, contrast, saturation, entropy*, and *colorfulness* [39]. Similar to color palette, these help with encoding low-level image representations into the model and enable more steerable generation.

**Text:** Large language models (LLMs) trained on large text corpora learn strong representations as shown by several works [24, 71, 89, 67]. We include *captions* from CC12M [16] and COYO700M [12]

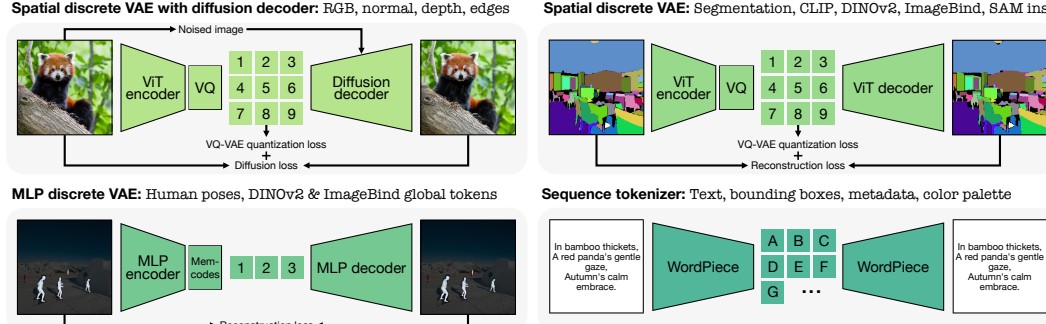

Figure 3: **Tokenization overview.** We employ suitable tokenization schemes for different modalities based on their format and performance. For image-like modalities and feature maps, we use spatial VQ-VAEs [66] with optional diffusion decoders for detail rich modalities like RGB. For non-spatial modalities like global tokens or parameterized poses, we compress them to a fixed number of discrete tokens using Memcodes [62] with MLP encoders and decoders. All sequence modalities are encoded as text using WordPiece [24]. The shown examples are real tokenizer reconstructions. Notice the low reconstruction error. See appendix D for more details and Fig. 13 for visualizations.

datasets, as well as web text from C4 [71] for language modeling. Next, we employ both a standard WordPiece [24] tokenizer for captions as [65] as well as *caption embeddings* obtained from a T5-XXL [71] encoder to capture better text representations, which have been shown to improve text-to-image generation fidelity [80, 14] (See fig. 4).

## 2.2 Tokenization

Tokenization consists of converting modalities and tasks into sequences or sets of *discrete tokens*, thereby unifying their representation space. This is critical for training large multimodal models as it confers the following key benefits: 1) It enables training multimodal and multitask models with a single pre-training objective. After tokenization, all tasks are formulated as a per-token classification problem using the cross-entropy loss. This improves training stability, enables full parameter sharing, and removes the need for task-specific heads, loss functions, and loss balancing. 2) It makes generative tasks more tractable by allowing the model to iteratively predict tokens, either autoregressively [72, 96] or through progressive unmasking [15, 14]. 3) It reduces computational complexity by compressing dense modalities like images into a sparse sequence of tokens. This decreases memory and compute requirements, which is crucial when scaling up to larger dataset and model sizes.

We use different tokenization approaches to discretize modalities with different characteristics. See fig. 3 for an overview. To summarize, we mainly use three different types of tokenizers, as explained below. Please see appendices D and H for more details and insights on tokenizer design choices.

**ViT tokenizer (with optional diffusion decoder):** We trained modality-specific ViT [25] based VQ-VAE [66] tokenizers for image-like modalities such as edges and feature maps. The resulting tokens form a small grid of size $14 \times 14$ or $16 \times 16$, according to the pseudo-labeler patch size. The edge tokenizers use a diffusion decoder [82, 65] to get visually more plausible reconstructions.

**MLP tokenizer:** For human poses and global embeddings from DINOv2 and ImageBind, we use Bottleneck MLP [6] based discrete VAEs with Memcodes quantization [62] to tokenize them into a small number of tokens, e.g. 16.

**Text tokenizer:** We leverage a WordPiece [24] tokenizer which is used to encode not only text, but also other modalities such as bounding boxes, color palettes and metadata using a shared set of special tokens to encode their type and values (See appendix D.6 for details).

## 2.3 Training details

**Datasets:** We perform the training in two stages, namely a 4M pre-training stage on a significantly larger image dataset, followed by a fine-tuning phase on a smaller dataset containing a larger number of modalities. Since the `4M-XL` model showed signs of overfitting on sequence modalities when trained on CC12M [16], we re-trained the models on COYO700M [12], containing 50 times more samples. COYO700M was pseudo labeled with the same modalities used for 4M. To cut down

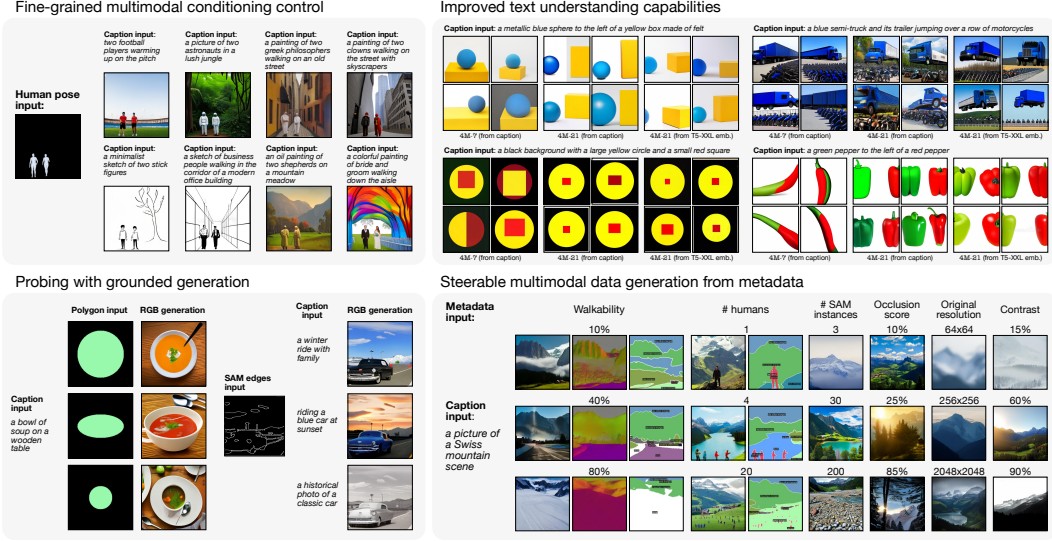

Figure 4: **Fine-grained & steerable multimodal generation. Top left:** 4M-21 can generate variants of images that are grounded in any input modality, here human poses. **Bottom left:** This enables us to perform multimodal edits (e.g. editing the shape of a polygon or grounding generation with edges) and probe the learned representation. For example, by only changing the shape of the ellipse, 4M-21 renders the bowl from different angles. **Top right:** By pre-training on 21 types of modalities, including T5-XXL embeddings, and *co-training with language modeling* on a large text corpus, we show improved text understanding capabilities (even when the input is captions instead of language model embeddings). **Bottom right:** Compared to generating images from captions only, metadata provides a more direct and steerable way of controlling the multimodal data generation process, enabling exciting further research into generative dataset design.

on pseudo labeling cost when expanding the number of modalities, we decided to pseudo label CC12M instead of COYO700M, and fine-tune the models with both new and old modalities. To avoid overfitting the larger models, we co-train them with samples from COYO700M. In addition to the previously mentioned multimodal datasets, we also included the C4 [71] text corpus in training. We perform the training by randomly sampling elements of each batch from any of these datasets, given a pre-determined set of sampling weights, and perform language modeling on them. Exact details on the training mixture are given in appendix E.2.

**Architecture:** We adopt 4M's encoder-decoder based transformer architecture with additional modality embeddings to accommodate new modalities. Similar to 4M, besides RGB tokens, the encoder directly accepts RGB pixels with a learnable patch-wise projection to enable use as a ViT [25] backbone for transfer learning.

**Masking strategy:** We used both multimodal random [7, 65] and span masking [71] strategies that mask input and target tokens. We invoke dataset mixing ratios and Dirichlet sampling parameters, $\alpha$, to ensure stable training on multiple modalities and datasets, as detailed in appendix E.2.

## 3 Multimodal capabilities

We demonstrate a broad range of capabilities unlocked by 4M-21, including steerable multimodal generation (Sec. 3.1), multimodal retrieval (Sec. 3.2) and strong out-of-the-box capabilities (Sec. 3.3). Please see the project website for more visualizations demonstrating these capabilities.

### 3.1 Steerable multimodal generation

4M-21 can predict any training modality by iteratively decoding tokens [65, 15, 14]. This is shown in fig. 2 where we can generate all modalities from a given input modality in a consistent manner. Furthermore, as we can generate *any* of the training modalities from *any* subset of other modalities, both conditionally and unconditionally, it enables several ways to perform fine-grained and multimodal generation, as shown in fig. 4. This includes diverse capabilities such as performing multimodal edits, probing the learned representations, and steering multimodal data generation.

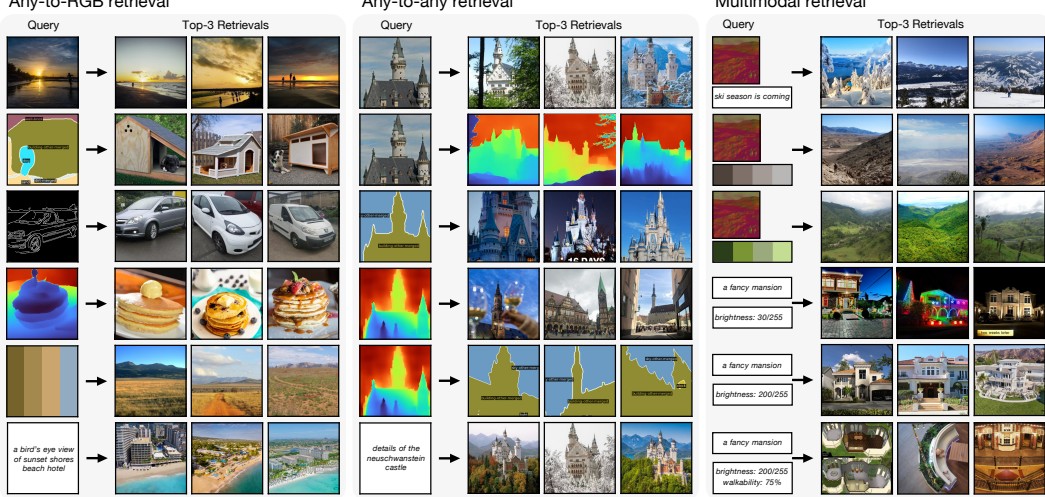

Figure 5: **Different modes of multimodal retrieval.** We perform *multimodal* retrievals by predicting global embeddings (here shown for DINOv2) from a given input (of any modality) using 4M-21 and comparing the cosine distances between the query and retrieval set embeddings. **Left:** Retrieving RGB images from distinctly different query modalities (here RGB, segmentation map, edges, depth map, color palette, and caption). **Middle:** Retrieving any modality using any other modality as the query input. Each query modality constrains the retrievals differently, e.g. here the RGB image and caption queries always yield Neuschwanstein castle retrievals. In contrast, for depth and semantic queries, the scene is more ambiguous, thus they retrieve other buildings with similar characteristics. **Right:** We can also *combine any subset of modalities* to define the query input, e.g. surface normals and a color palette, to better control the retrieval. See appendix B.2 for more results.

Moreover, 4M-21 exhibits improved text understanding capabilities leading to geometrically and semantically plausible generations, both when conditioning on T5-XXL embeddings and on regular captions (fig. 4, top right). Please see appendix I.2 for additional results.

## 3.2 Multimodal retrieval

Our model can also perform multimodal retrievals by predicting global embeddings of DINOv2 and ImageBind *from any (subset) of the input modalities*. Once the global embeddings are obtained, the retrieval is done by finding the retrieval set samples with the smallest cosine distance to the query [68, 35]. As shown in fig. 5, this unlocks retrieval capabilities that were not possible with the original DINOv2 and ImageBind models such as retrieving RGB images or any other modality via using any other modality as the query. Furthermore, one can combine multiple modalities to predict the global embedding, resulting in better control over retrievals, as shown on the right. Please see appendix I.3 for additional results.

## 3.3 Evaluating out-of-the-box capabilities

4M-21 is capable of performing a range of common vision tasks out-of-the-box, as demonstrated visually in fig. 6. In table 1, we evaluate the performance on DIODE [90] surface normal and depth estimation, COCO [57] semantic and instance segmentation, 3DPW [91] 3D human pose estimation, and do ImageNet-1K [79] kNN retrieval using predicted DINOv2 global tokens. We compare against the pseudo labeling networks, strong baselines, and the 4M model from [65] trained on 7 modalities. For surface normal estimation and semantic segmentation, we observed that ensembling multiple predictions significantly improves performance, see appendix F for more details and results.

Our model consistently achieves strong out-of-the-box performance, and often matches or even outperforms the pseudo labelers and other specialist baselines, *while being a single model for all tasks*. Notice the large performance gap with other multitask models like Unified-IO [61] and Unified-IO-2 [60]. For kNN retrieval, 4M-21 XL performance approaches the tokenizer bound, i.e. the retrieval performance using the DINOv2 tokenizer reconstructions. While the smaller models lag behind 4M models, we observe that 4M-21 XL is able to match the performance of 4M-7 XL, while being trained to solve three times more tasks. The trend over the model size needing to be larger is expected as the number of tasks increase.

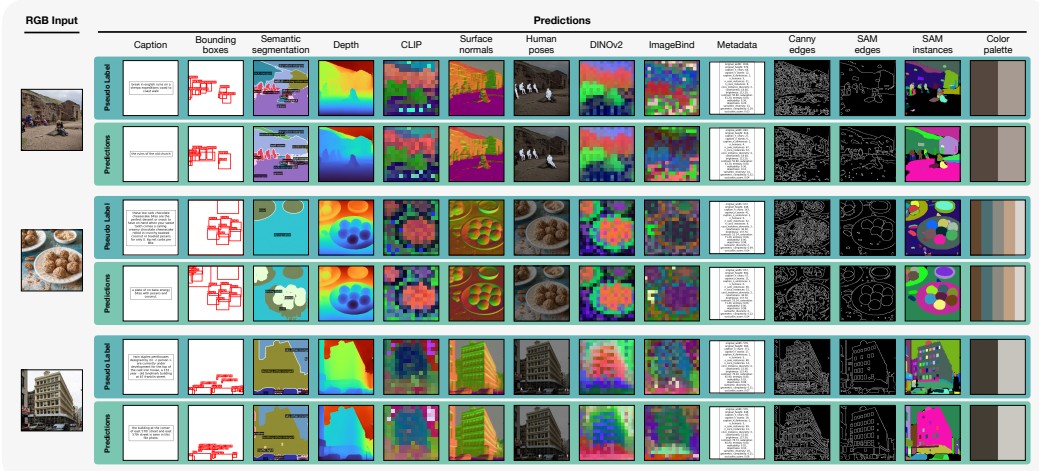

**Figure 6: Out-of-the-box vision tasks.** Given an RGB image, `4M-21` can predict all tasks successfully, as can be seen from their high consistency with the pseudo labels. See fig. 7 for more results.

**Table 1: Out-of-the-box (zero-shot) performance.** We show the performance for a common subset of tasks: surface normals and depth estimation on DIODE [90], semantic and instance segmentation on COCO [57], k-NN retrieval on ImageNet-1K [79], and 3D human keypoint estimation on 3DPW [91]. We compare to a set of strong baselines and specialist models, including our pseudo labelers. The model learned to solve all the tasks without a loss of performance, is significantly better than the baselines, and is competitive with pseudo labelers, *while being a single model for all tasks*. Compared to `4M-7`, the `4M-21` model preserved its performance while solving 3x more tasks. ✗ denotes that a given model cannot solve the task out-of-the-box. * shows the tokenizer reconstruction quality and provides an estimate on the performance upper bound due to tokenization. See fig. 14 for qualitative comparisons. Best results are **bolded**, second best underlined.

| | Method | Normals ↓ | Depth ↓ | Sem. seg. ↑ | Inst. seg. ↑ | IN1K kNN ↑ | 3D human KP ↓ |
|---|---|---|---|---|---|---|---|
| Pseudo labelers | Omnidata [46] | 22.5 | **0.68** | ✗ | ✗ | ✗ | ✗ |
| | M2F-B [20] | ✗ | ✗ | 45.7 | ✗ | ✗ | ✗ |
| | SAM [49] | ✗ | ✗ | ✗ | 32.9 | ✗ | ✗ |
| | DINOv2-B14 [68] | ✗ | ✗ | ✗ | ✗ | 82.1 / 93.9 | ✗ |
| | ImageBind-H14 [35] | ✗ | ✗ | ✗ | ✗ | 81.1 / **94.4** | ✗ |
| | 4D-Humans [37] | ✗ | ✗ | ✗ | ✗ | ✗ | **81.3** |
| | OASIS [19] | 34.3 | ✗ | ✗ | ✗ | ✗ | ✗ |
| | MiDaS DPT [73] | ✗ | 0.73 | ✗ | ✗ | ✗ | ✗ |
| | M2F-S [20] | ✗ | ✗ | 44.6 | ✗ | ✗ | ✗ |
| | M2F-L [20] | ✗ | ✗ | 48.0 | ✗ | ✗ | ✗ |
| | HMR [45] | ✗ | ✗ | ✗ | ✗ | ✗ | 130.0 |
| | UnifiedIO-B [61] | 35.7 | 1.00 | 32.9 | ✗ | ✗ | ✗ |
| | UnifiedIO-L [61] | 33.9 | 0.87 | 41.6 | ✗ | ✗ | ✗ |
| | UnifiedIO-XL [61] | 31.0 | 0.82 | 44.3 | ✗ | ✗ | ✗ |
| | UnifiedIO 2-L [60] | 37.1 | 0.96 | 38.9 | ✗ | ✗ | ✗ |
| | UnifiedIO 2-XL [60] | 34.8 | 0.86 | 39.7 | ✗ | ✗ | ✗ |
| | UnifiedIO 2-XXL [60] | 37.4 | 0.84 | 41.7 | ✗ | ✗ | ✗ |
| | 4M-7 B [65] | 21.9 | 0.71 | 43.3 | ✗ | ✗ | ✗ |
| | 4M-21 B | 21.7 | 0.71 | 42.5 | 15.9 | 73.1 / 89.7 | 108.3 |
| | 4M-7 L [65] | 21.5 | 0.69 | 47.2 | ✗ | ✗ | ✗ |
| | 4M-21 L | 21.1 | 0.69 | 46.4 | 31.2 | 77.0 / 91.9 | 97.4 |
| | 4M-7 XL [65] | **20.6** | 0.69 | **48.1** | ✗ | ✗ | ✗ |
| | 4M-21 XL | 20.8 | **0.68** | **48.1** | 32.0 | 78.3 / 92.4 | 92.0 |
| | Tokenizer bound* | 4.0 | 0.06 | 90.5 | 91.2 | 80.2 / 93.0 | 17.5 |

## 4 Transfer experiments

To study the scaling characteristics of pre-training any-to-any models on a larger set of modalities, we train models across three different sizes: B, L, and XL. We then transfer their encoders to downstream tasks and evaluate on both unimodal (RGB) and multimodal (RGB + Depth) settings. The decoders are discarded for all transfer experiments, and we instead train task-specific heads. We perform self-comparisons in a similar manner to [65, 7], as well as comparing to a set of strong baselines.

**Unimodal transfers.** For unimodal transfers we leverage the RGB patch embeddings learned during the pre-training, as RGB pixel inputs are used alongside the tokenized modalities. For the XL models

Table 2: **Unimodal transfer study.** We transfer 4M-21 and baselines to ImageNet-1K [79] classification, ADE20K [106] semantic segmentation, NYUv2 [84] depth estimation, and ARKitScenes [9] (ARKS) 3D object detection. We observe that 4M-21 **1)** does not lose performance for the transfer tasks that are similar to the seven modalities of 4M, i.e. first three columns of the results, while being able to solve many more, and **2)** leads to improved performance for novel tasks that are more different from 4M modalities, e.g. 3D object detection (last column). The improvements are further verified in the multimodal transfer results (Table 3) showing the usefulness of new modalities. Best results per task are **bolded**, second best underlined.

| Method | Pre-training data | Enc. param. | IN1K Acc.↑ | ADE20K mIoU↑ | NYUv2-D $\delta_1$ acc.↑ | ARKS AP$^{3D}$↑ |
|---|---|---|---|---|---|---|
| MAE B [40] | IN1K | | 84.2 | 46.1 | 89.1 | 30.9 |
| DeiT III B [88] | IN21K | | 85.4 | 49.0 | 87.4 | 36.1 |
| MultiMAE B [7] | IN1K | | 84.0 | 46.2 | 89.0 | 34.2 |
| DINOv2 B [68] | LVD142M | 86M | 85.3 | 51.6 | 92.2 | 38.1 |
| 4M-7 B [65] | CC12M | | 84.5 | 50.1 | 92.0 | 40.3 |
| 4M-7 B (Ours) | COYO | | 84.4 | 49.4 | 91.4 | 38.6 |
| 4M-21 B | CC12M+COYO+C4 | | 84.5 | 50.1 | 90.8 | 42.4 |
| MAE L [40] | IN1K | | 86.8 | 51.8 | 93.6 | 36.2 |
| DeiT III L [88] | IN21K | | 87.0 | 52.0 | 89.6 | 40.3 |
| DINOv2 L [68] | LVD142M | 303M | 86.7 | 53.4 | 94.1 | 42.8 |
| 4M-7 L [65] | CC12M | | 86.6 | 53.4 | 94.4 | 46.8 |
| 4M-7 L (Ours) | COYO | | 86.7 | 53.5 | 94.3 | 45.2 |
| 4M-21 L | CC12M+COYO+C4 | | 86.5 | 53.4 | 93.7 | 47.0 |
| DINOv2 g [68] | LVD142M | 1.1B | **88.0** | **58.7** | 92.5 | 45.3 |
| 4M-7 XL [65] | CC12M | | 87.0 | 55.0 | 96.1 | 48.1 |
| 4M-7 XL (Ours) | COYO | 1.2B | 87.1 | 56.1 | **96.5** | 47.3 |
| 4M-21 XL | CC12M+COYO+C4 | | 87.1 | 56.0 | **96.5** | **48.4** |

and DINOv2 g, we perform parameter-efficient fine-tuning using LoRA [42] instead of full fine-tuning, which significantly improves results for XL models. We did not observe similar performance gains for the smaller models. Further training details are described in appendix G.

We evaluate on ImageNet-1K classification [23, 79], ADE20K semantic segmentation [106], NYUv2 depth estimation [84], and ARKitScenes [9] 3D object detection tasks. Some transfer tasks are completely unseen during pre-training, e.g. object classification or 3D object detection, while others are included as different instantiations, e.g. absolute depth instead of relative depth, or using ADE20K instead of COCO classes. We follow the best practices and commonly used settings from other papers [65].

The results are shown in table 2. We make the following observations: **1)** for the transfer tasks that are similar to the seven modalities of 4M, e.g. semantic segmentation or depth, 4M-21 does not lose performance due to being trained on many more modalities, **2)** for novel transfer tasks like 3D object detection that are sufficiently different from 4M modalities, we observe an improved performance. Moreover, the performance improves with larger model sizes, showing promising scaling trends. These trends can be further seen in the multimodal transfer results, which we explain next.

**Multimodal transfers.** We perform multimodal transfers on NYUv2, Hypersim [75] semantic segmentation, and 3D object detection on ARKitScenes. We compare transfers using RGB images only, and RGB pixels + tokenized sensory depth as inputs. As table 3 shows, 4M-21 makes strong use of optionally available depth inputs and significantly improves upon the baselines.

Table 3: **Multimodal transfer study.** We transfer both 4M-21 and 4M (pre-trained on CC12M) to NYUv2 and Hypersim segmentation, and 3D object detection on ARKitScenes. All models are able to use optionally available depth when it is of high quality (Hypersim & ARKitScenes), while our model achieves the best results. Best results are **bolded**, second best underlined.

| Method | NYUv2-S mIoU ↑ | | Hypersim mIoU ↑ | | ARKitScenes AP$^{3D}$↑ | |
|---|---|---|---|---|---|---|
| | RGB | RGB-D | RGB | RGB-D | RGB | RGB-D |
| 4M-7 B | 56.6 | 57.5 | 40.2 | 43.9 | 40.3 | 46.5 |
| 4M-21 B | 58.7 | 59.7 | 38.6 | 46.4 | 42.4 | 48.1 |
| 4M-7 L | 61.2 | 61.4 | **48.7** | 50.5 | 46.8 | 49.5 |
| 4M-21 L | 61.8 | 61.8 | 47.3 | 50.7 | 47.0 | 50.1 |
| 4M-7 XL | 62.1 | 61.2 | 48.6 | 51.0 | 48.1 | 50.1 |
| 4M-21 XL | **63.9** | **63.9** | 48.6 | **52.5** | **48.4** | **51.3** |

## 5 Related Work

Multitask learning in vision involves training a single model to perform multiple visual tasks efficiently [13, 78]. Earlier methods [28, 64, 50] combined multiple dense vision tasks into a single model but faced challenges scaling to a larger variety

of tasks and modalities, limited by training instabilities and the need for careful task selection and loss balancing to reduce negative transfer [48, 102, 86, 100].

Recently, discrete tokenization has enabled a shift towards integrating numerous vision tasks into unified multimodal and multitask models such as Gato [74], OFA [93], Pix2Seq [17, 18], UnifiedIO [61, 60], 4M [65], and more [51, 83, 5, 41, 26, 44, 4, 97, 87, 108, 54, 105]. These methods first transform various modalities and tasks into sequences or sets of discrete tokens [24, 52, 66, 31, 95], and then train a single Transformer on these tokens using either a sequence modeling [74, 93, 61, 17, 18, 51, 83] or masked modeling objective [65, 94]. Some methods (e.g. Gato [74], UnifiedIO [61, 60]) perform co-training on multiple disjoint datasets and are capable of performing a wide range of tasks, but not jointly. In contrast, methods like 4M [65] train on a single aligned dataset through the use of pseudo labeling, enabling any-to-any modality prediction but on a typically more limited set of modalities. We significantly expand upon them by adding the ability to use this framework for an even greater amount of modalities and capabilities.

Furthermore, masked modeling has proven effective for learning useful representations in both NLP [24, 71] and vision [40, 8, 107, 29]. Extending it to multimodal domains [7, 36, 94, 65] enables strong cross-modal representations which is critical for multimodal learning. When combined with tokenization, masked modeling also enables generative applications [15, 55, 14, 85, 65]. Our work highlights the ability of masked modeling to expand to a much greater set of modalities than previously shown, improving upon the out-of-the-box and multimodal generation capabilities of previous works.

## 6    Limitations and Discussion

We developed an any-to-any model on tens of diverse modalities and tasks. This was achieved by mapping all modalities to discrete sets of tokens via modality-specific tokenizers and using a multimodal masked training objective [65]. We successfully scaled the training to 3 billion parameters and to 21 modalities and different datasets, without a degradation in performance compared to the existing expert single/few task models. This results in strong out-of-the-box capabilities as well new potential for multimodal interaction, generation, and retrieval, all by a single unified model. Below, we discuss limitations and future work.

*Transfer/emergent capabilities:* One hope from training a single network on several tasks is leading to a model that can solve novel tasks, often referred to as "transfer" or "emergent" capabilities. While, as we showed, a multitask model brings several key advantages even without transfer/emergence (e.g., efficiency, using a single model for broad out-of-the-box capabilities, modality fusion, etc.), we observe that **the potential for transfer/emergence remains largely untapped**. In general, compared to LLMs, vision/multimodal models have not shown exciting results in terms of transfer/emergence yet. We find this to be an important point to address in the future, e.g., via designing multitask architectures that have emergence, in contrast to out-of-the-box capabilities, as their main objective.

*Better tokenization:* Like any token-based model, ours can directly benefit from progress on tokenizers, e.g. higher reconstruction fidelity.

*Co-training on partially aligned datasets:* We showed the possibility of training on partially aligned datasets, e.g. text data from C4 and other modalities from CC12M, yet further investigations and a larger mixture of datasets are expected to bring stronger capabilities.

**Acknowledgement:** This work was supported as part of the Swiss AI initiative by a grant from the Swiss National Supercomputing Centre (CSCS) under project ID a08 on Alps. We also acknowledge the support of the ETH4D and EPFL EssentialTech Centre Humanitarian Action Challenge Grant.

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

# Appendix

# A  Code, Pre-trained Models & Interactive Visualizations

Please see our website for documented open-source code, pre-trained model *and* tokenizer weights, as well as an overview video and additional interactive visualizations.

# B  Multimodal Capabilities

## B.1  Additional multimodal generation & probing visualizations

Please see Figures 7, 8, 9, 10 for additional qualitative results on any-to-any generation, controlled generation, and text understanding capabilities of our model.

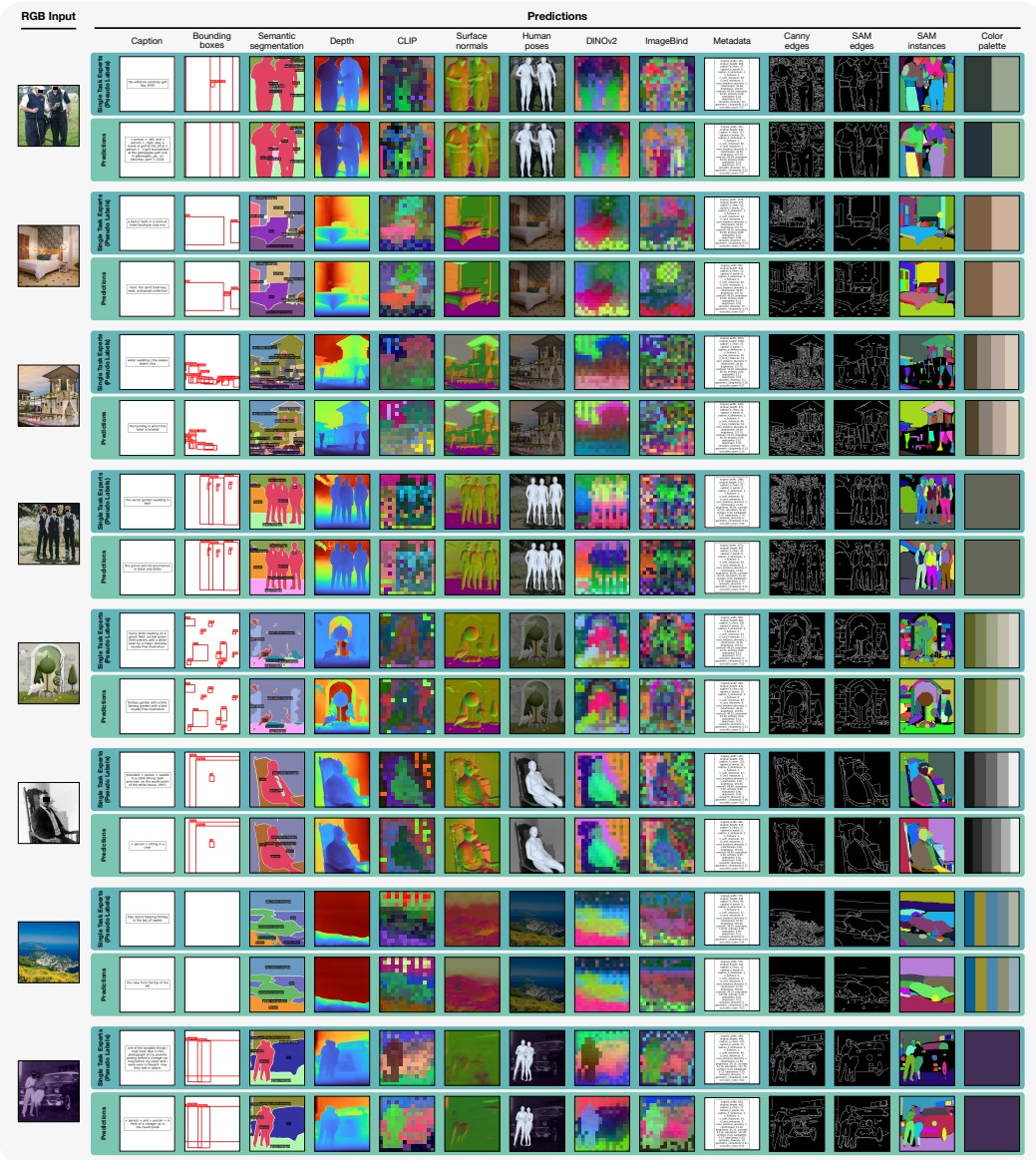

Figure 7: **RGB-to-any generation.** This is an extension of fig. 6 and visualizes the model's out-of-the-box capabilities on various vision tasks, compared to the pseudo labeler outputs.

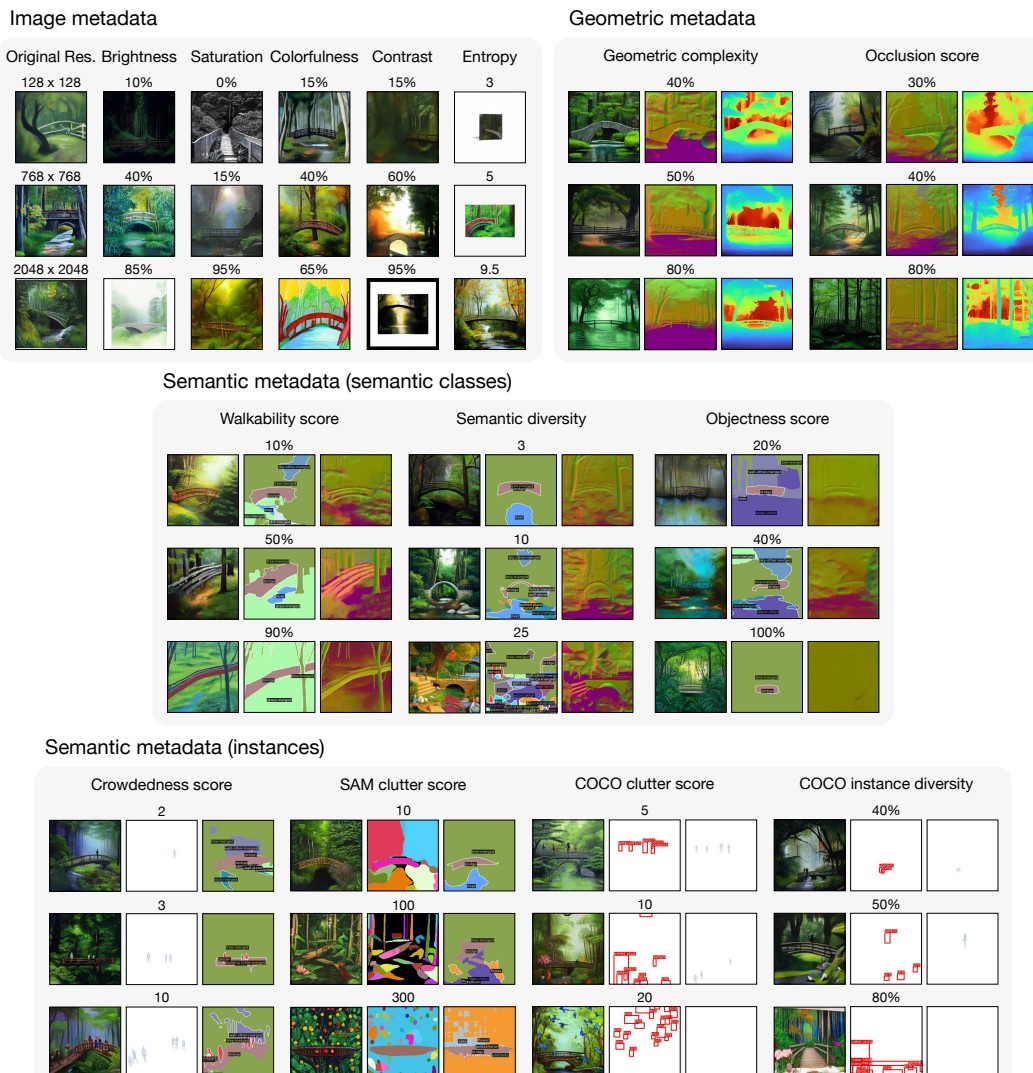

Figure 8: **Steerable multimodal generation using metadata.** This is an extension of fig. 4 and shows our model's capability of generating multimodal data by conditioning on a wide set of controls. The common caption for all examples is *"a painting of a bridge in a lush forest"*.

Varying SAM polygon instances

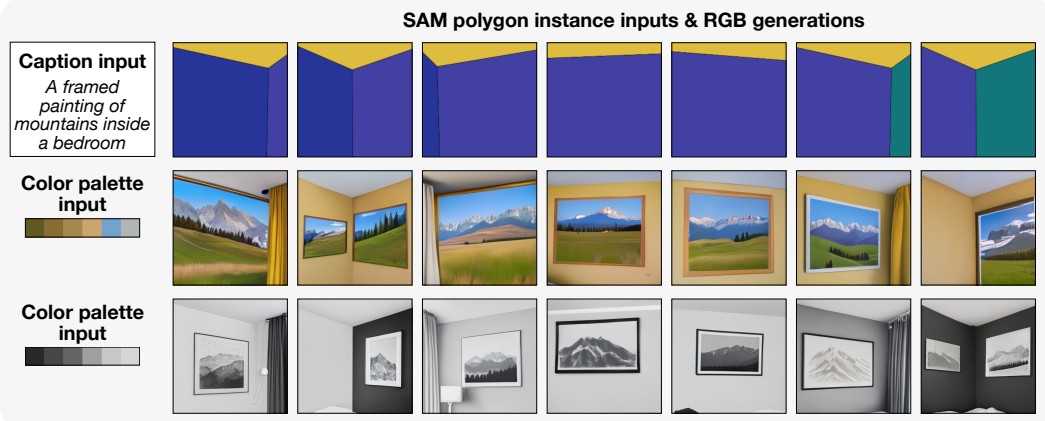

Varying color palette

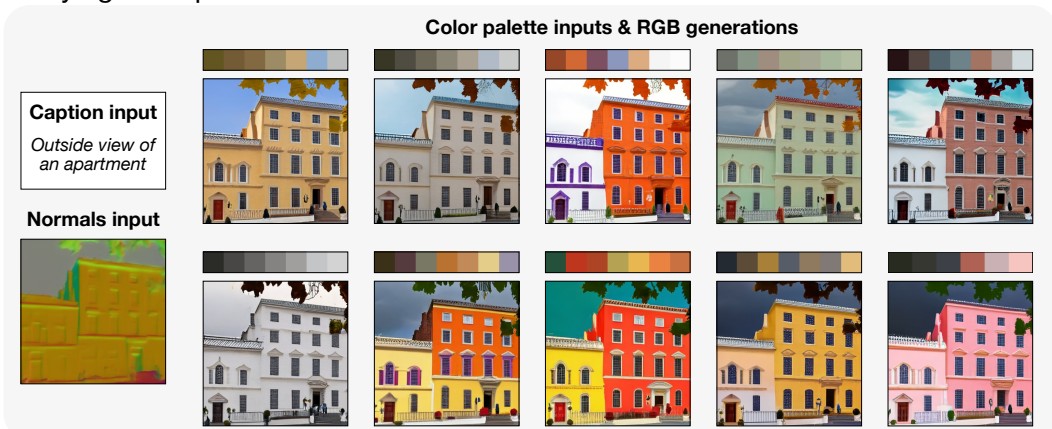

Figure 9: **Probing with grounded generation.** This is an extension of fig. 4 and further shows our model's capability on performing generation by conditioning on multimodal input. The top row varies SAM instances and combines them with a fixed caption and color palette input. The bottom row fixes the normals and caption inputs and varies the color palette.

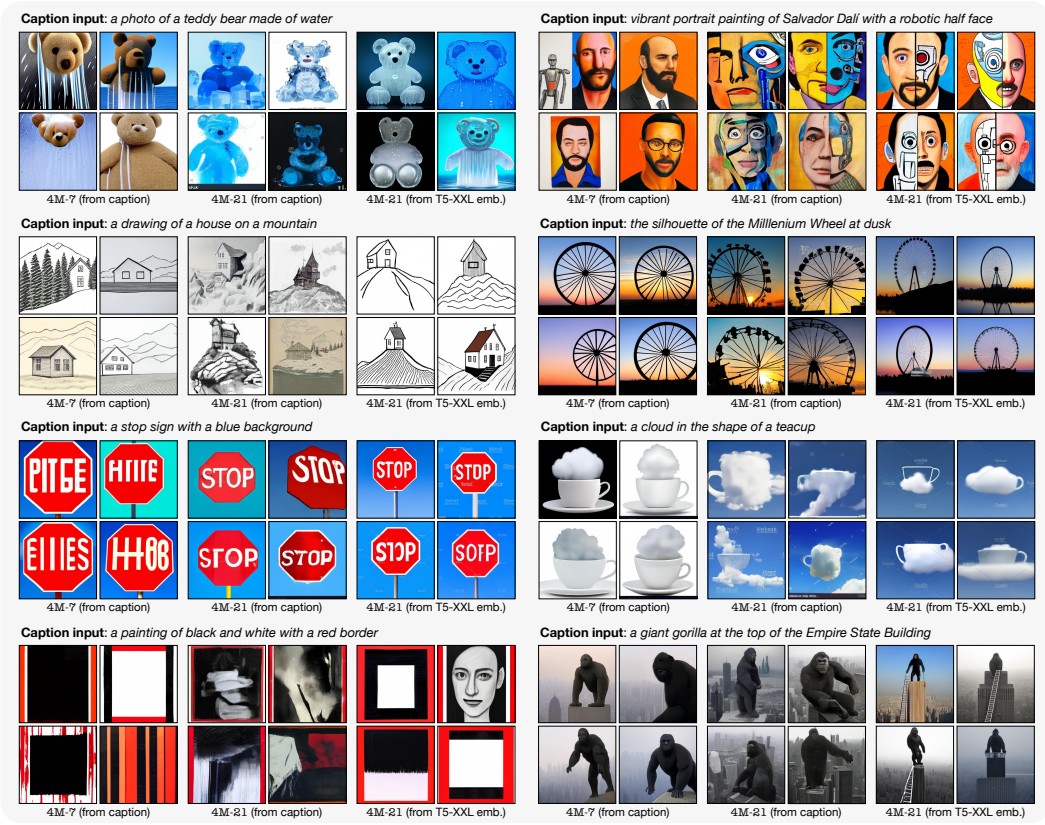

Figure 10: **Text understanding.** This is an extension of fig. 4 and further demonstrates improved text understanding capabilities of our method compared to 4M for several caption inputs.

## B.2 Additional retrieval visualizations

Please see Figures 11 and 12 for additional qualitative results on RGB-to-Any and Any-to-RGB retrievals.

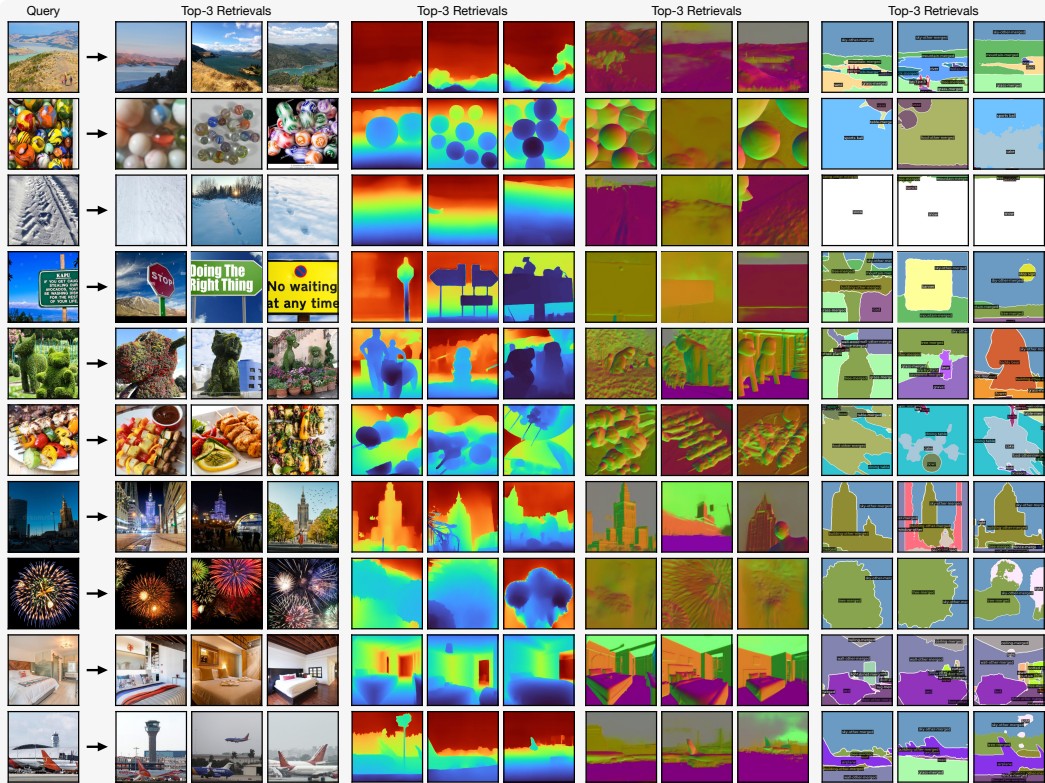

Figure 11: **RGB-to-Any retrieval.** This is an extension of fig. 5 and further demonstrates cross-modal retrieval capabilities of our model. Here our model successfully retrieves several modalities (RGB, depth, normals, segmentation) using the RGB image as the query input.

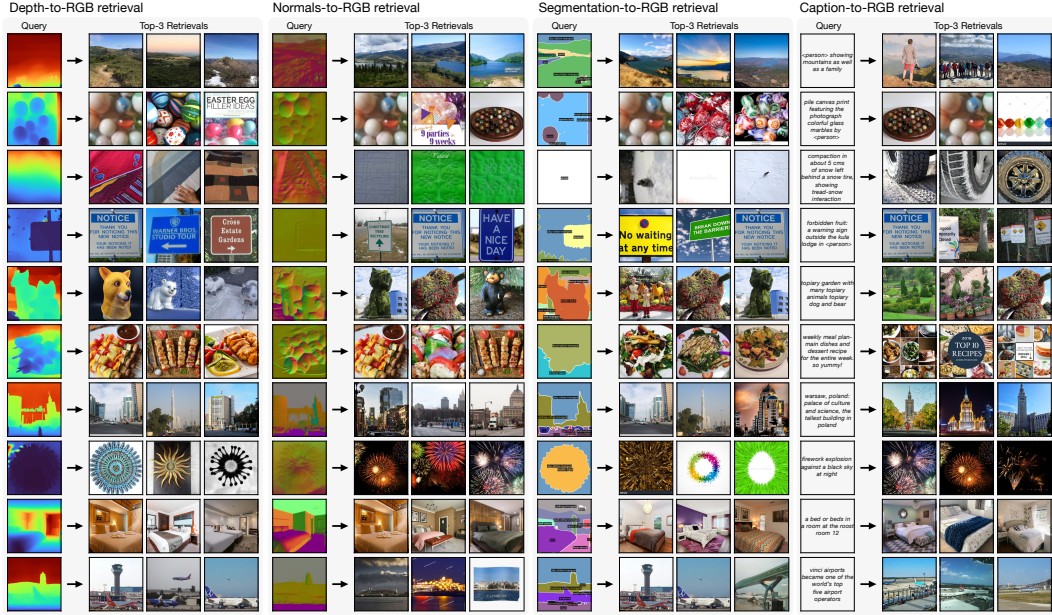

Figure 12: **Any-to-RGB retrieval.** This is an extension of fig. 5 and further demonstrates cross-modal retrieval capabilities of our model. Here our model successfully retrieves RGB images when the query inputs are from depth, normals, segmentation, and caption modalities.

## C Additional Ablations

### C.1 Ablation of pre-training data and modalities

For training 4M-21, we initialize the training using 4M models that we pre-trained on COYO700M [12]. We ablate in Table 4 different choices of training data and modalities. We can see that performing co-training on C4 [71] and COYO700M [12] has the potential to slightly improve transfer performance on average.

Table 4: **Pre-training data and modality mixture ablation:** We ablate different pre-training modality and dataset choices on B models. * represents the models initialized from the corresponding 4M models trained on COYO700M.

| Method | Pre-training data | ImageNet-1K Top-1 acc. ↑ | ADE20K mIoU ↑ | NYUv2 depth $\delta_1$ acc. ↑ | ARKitScenes $AP^{3D}$ ↑ |
|---|---|---|---|---|---|
| 4M-7 B [65] | CC12M | 84.5 | **50.1** | **92.0** | 40.3 |
| 4M-7 B | COYO700M | 84.4 | 49.4 | 91.4 | 38.6 |
| 4M-7 B * | CC12M | 84.5 | 49.2 | 91.0 | 39.5 |
| 4M-21 B * | CC12M | 84.4 | 49.2 | 90.9 | 40.0 |
| 4M-21 B * | CC12M+C4 | **84.6** | 49.5 | 90.4 | 41.2 |
| 4M-21 B * | CC12M+COYO700M+C4 | 84.5 | **50.1** | 90.8 | **42.4** |

### C.2 Ablation of ensembling the predictions

Unlike the deterministic pseudo labeler and other state of the art networks we compared against in table 1, our model can produce multiple prediction given the same RGB input through repeated sampling with a different seed. As shown in Table 5, ensembling ten samples of predicted surface normals and semantic segmentation maps can significantly improve the reported metrics. While ensembling improves upon these metrics, we note that the ensembled predictions can be comparatively blurrier around object edges than any individual sample.

Table 5: **Ensembling ablation:** We ablate ensembling multiple predictions on DIODE normals and COCO semantic segmentation compared to no ensembling. As the results suggest, ensembling in all cases improves the quantitative results.

| Method | DIODE Normals mean angle error ↓ | | COCO Semseg mean IoU ↑ | |
| --- | --- | --- | --- | --- |
| | No Ensemble | Ensemble | No Ensemble | Ensemble |
| 4M-21 B | 22.3 | **21.7** | 39.0 | **42.5** |
| 4M-21 L | 21.7 | **21.1** | 43.8 | **46.4** |
| 4M-21 XL | 21.3 | **20.8** | 46.5 | **48.1** |

# D   Multimodal Dataset & Tokenization Details

## D.1   Pseudo labeled multimodal training dataset

Similar to 4M, to have an aligned multimodal dataset, we pseudo label the CC12M dataset using strong specialized models for each task. The pseudo labeling of existing modalities is done in the same fashion as 4M, using Omnidata DPT-Hybrid [73] for surface normals and depth estimation, COCO Mask2Former [20] with a SwinB [58] backbone for semantic segmentation, COCO ViTDet ViT-H model [56] initialized from MAE weights [40] for bounding boxes, and CLIP-B16 [70] with ViT-B/16 visual backbone backbone for CLIP feature maps.

**3D human poses.** We use 4D-Humans [37] to extract 3D pose and shape parameterized by an SMPL model. For the images in CC12M without humans, we set the pose label to a "none" token. For the images with humans, we form a sequence by concatenating the bounding box, body pose, camera, and shape values in a sequence for each human instance. As data augmentation, we randomly shuffle the order of each component in the sequence.

**SAM instances.** Besides semantic segmentation and bounding boxes, SAM [49] instance segmentation also provides some level of semantic information from an image by clustering together semantically similar pixels in it. Unlike semantic segmentation, SAM instances are not restricted to a specific set of classes and can segment in more detail. We use the SAM H model and query it with points in a grid format to obtain the instances. We also considered the SAM-HQ [47] H model, however in the grid-point querying format, it yields very similar results to SAM. We found $32 \times 32$ query points to be the optimal choice both for pseudo labeling speed and quality.

**DINOv2 and ImageBind global features & feature maps.** We extract both dense feature maps and global embeddings, i.e. cls token embeddings, from DINOv2-B14 [68] and ImageBind-H14 [35] pre-trained models. For the latter, we only extracted the image embeddings, incorporating other modality embeddings such as thermal or audio could be interesting future work.

**T5-XXL embeddings.** Language model embeddings, such as from T5-XXL [71], have been shown to improve the generation fidelity and text understanding capabilities of text-to-image generative models [80, 14]. Consequently, we use the T5-XXL encoder to extract text embeddings from all CC12M captions, without any preprocessing of the text. Unlike other modalities, we do not convert these text embeddings to a sequence of discrete tokens or treat them as targets (similar to the RGB pixel modality variant). Instead, we only provide them as inputs using a linear projection from the T5-XXL embedding dimension ($d_{\text{T5-XXL}} = 4096$) to our model's embedding dimension.

**Image metadata.** From RGB images, we directly extract different types of metadata like the *original height and width* before cropping [69], *brightness, contrast, saturation* and *entropy*. We additionally extract a notion of *colorfulness*, following [39].

**Semantic metadata.** We compute the *crowdedness score* as the number of humans in the pseudo labeled human poses, the *SAM clutter score* as the number of SAM instances, the *COCO clutter score* as the number of COCO instances, the *COCO instance diversity* as the number of unique COCO instance classes, and the *semantic diversity* as the number of unique COCO semantic classes in an image. For *caption length*, we count the number of characters, words, and sentences. As *objectness score*, we count the percentage of pixels in the COCO semantic segmentation map that belong to countable classes (indices 87, 90, 96, 97, 98, 99, 100, 101, 102, 103, 105, 106, 109, 110,

111, 112, 113, 117, 118, 119, 122, 123, 124, 125, 126, 129, 131, 132), and for the *walkability score* we count classes such as 'road' (indices 87, 90, 97, 100, 102, 105, 106, 122, 123, 125, 126, 132).

**Geometric metadata.** To compute the *occlusion score*, we first generate occlusion edges from depth images by applying a Sobel filter, followed by counting the percentage of occlusion edge pixels that surpass a threshold of 0.3. As a notion of *geometric complexity*, we project surface normal pixels onto the unit sphere, and compute their angular variance. Note that images of indoor scenes or caves featuring large surfaces pointing in all different directions receive a high score in this metric, while ones with a more localized geometric variance get a somewhat lower score. Exploring other potential notions of geometric complexity can be an interesting future addition.

**Color palette.** For every RGB image, we extract between one and seven color palettes using PyPalette [2]. During training, we randomly sample one of the color palettes to enable users to input palettes with different levels of granularity.

**SAM edges and canny edges.** Edges are a convenient way of grounding image generation on shapes contained in images [104]. To pseudo label edges, we apply the OpenCV canny edge detector on SAM instance maps and RGB, to obtain SAM edges and canny edges respectively.

## D.2 Tokenization of human poses

We use a BottleneckMLP [6] with 6 blocks and 1024 width to compress pose into 8 tokens. We use 1024 vocabulary size, and trained using smooth L1 loss for 15 epochs on CC12M training data. We also binned the global orientation, body shape, and bounding boxes into 1000 discrete bins similar to [17]. The final sequence is obtained by also adding identifiers, i.e. "bbox", "pose", "shape", before the corresponding sub-sequence.

## D.3 Tokenization of SAM instances

The SAM instance tokenizer is a ViT-based VQ-VAE that tokenizes $64 \times 64$ binary masks into 16 tokens using a vocabulary size of 1024. The tokenizer is trained using the cross-entropy loss for 24 epochs on CC12M training data, by resizing individual masks into a square aspect ratio image of $64 \times 64$ pixels. To preserve the SAM instances' original location, width, and height in the image, their bounding boxes are extracted. The final sequence for each instance is formed by appending the identifier "polygon" to 4 numbers that specify the bounding box of the instance, along with the 16 token IDs.

## D.4 Tokenization of global feature maps

Similar to human poses, we use BottleneckMLP with 6 blocks and 1024 width to compress DINOv2-B14 and ImageBind-H14 global embeddings into 16 tokens. We use 8192 vocabulary size, and trained using cosine similarity loss for 15 epochs.

## D.5 Tokenization of dense feature maps

We follow [65] and tokenize CLIP-B16, DINOv2-B14, and ImageBind-H14 dense feature maps into 196, 256, and 256 tokens, respectively, using a ViT-based VQVAE with 8192 vocabulary size and smooth L1 loss.

## D.6 Tokenization of sequence modalities

We tokenize text, color palette, metadata, and bounding boxes using a WordPiece tokenizer by fitting it on all captions and 4000 "special value" tokens, with a joint vocabulary size of 30k. These special tokens are divided into four groups, each with 1000 values, i.e. v0=0, v0=1, ..., v0=999, v1=0, v1=1, ... v1=999, v2=0, v2=1, ..., v2=999, v3=0, v3=1, ..., v3=999. For bounding boxes, we follow 4M [65] and represent xmin, ymin, xmax, ymax coordinates using v0, v1, v2, v3 tokens respectively. Other modalities are tokenized by binning their values into corresponding bins, e.g. color palette sequence is formed as $color = c \ R = r \ G = g \ B = b \ R = r, ...$ where $c$ takes a value between 1 and 7 and

specifies the number of colors in the palette and $r, g, b$ takes values between 0-255. We chose to model metadata using interleaved pairs of special tokens, where the first one specifies the type of metadata modality, and the second specifies its value. For example, a crowdedness score of 3 and a brightness of 120 would be specified as the sequence `v1=5 v0=3 v1=10 v0=120`. During training the number of metadata entries and their order is randomized. All of this results in a sequence prediction formulation, following [65, 17].

### D.7 Tokenization of Canny and SAM edges

We use a VQ-VAE with a diffusion decoder, similar to [65] to tokenize the edge modalities. We use the same tokenizer as it reconstructs both edges similarly well.

### D.8 Tokenizer reconstruction quality

We provide qualitative results for tokenizer reconstructions in fig. 13.

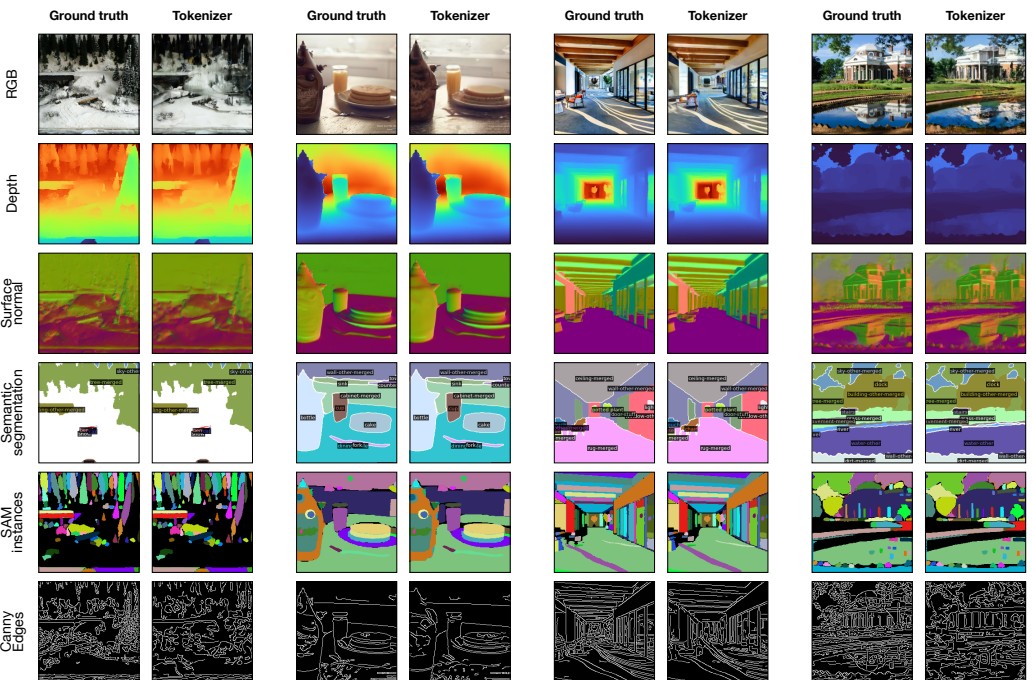

Figure 13: **Tokenizer reconstruction quality.** Our multimodal tokenizers can reconstruct the ground truth well. Here we show sample reconstructions of RGB, depth, surface normals, semantic segmentation, SAM instances, and canny edges on (pseudo) labels from the CC12M validation set at $224 \times 224$ resolution. Quantitative evaluations are provided in table 1 for different tasks and datasets (last row, *Tokenizer bound*), confirming the reconstruction quality.

## E   Training Details

Please see Tab. 6 for an overview of pre-training settings. For more accurate model comparisons, the architecture and overall training objective of our B, L, and XL models are the same as those of 4M models. However, we do modify and improve various aspects of the training process that allow us to significantly increase the number of training modalities. These changes concern modality-specific accommodations to the masking strategy, the ability to co-train on several datasets, and the use of a more diversified multimodal masking strategy. We describe these modifications below:

## E.1 Modality-specific accommodations

**Positional and modality embeddings.** As with 4M, 4M-21 incorporates both learnable modality embeddings and fixed sine-cosine positional embeddings for each modality. The positional embeddings are either 1D or 2D depending on the modality type.

**Metadata grouping and chunk-based masking.** To address the sparsity and number of different types of metadata, the metadata modalities are all grouped together as a single modality during training. This prevents the over-allocation of tokens to sparse metadata, enabling a more balanced distribution of the token budget across modalities. However, the standard span masking from T5 [71] and 4M [65] performs random uniform masking at the token level, which can lead to pre-training inefficiencies [53] and make conditioning on specific metadata difficult, as conditioning on just one of them would rarely occur during pre-training with this masking strategy. Instead, we propose to mask chunks of sequence (similar to PMI-Masking [53]), where the span masking is performed per chunk of metadata instead of at the token level.

## E.2 Multidataset co-training and diversified multimodal masking strategy

**Multi-dataset support.** Unlike 4M which was only trained on a single aligned dataset, we train 4M-21 on multiple datasets simultaneously. This flexibility allows for the inclusion of datasets with varying numbers of modalities, which enables training on both large-scale datasets with a smaller number of modalities and smaller datasets with a larger diversity of modalities.

**Sampling and masking strategies.** Our data sampling process involves selecting a training dataset based on its sampling weight, followed by choosing a masking strategy from the dataset-specific mixture of masking strategies. Input and target tokens are then sampled using the selected strategy.

**Co-training datasets.** We co-train on several datasets to improve the model's performance and the data diversity. These include CC12M [16], which comprises about 10 million text-image samples fully pseudo labeled with all 21 modalities, and accounts for 60% of our training samples. Additionally, we include COYO700M [12], with approximately 500 million text-image samples pseudo labeled with the 7 modalities of 4M, and accounts for 20% of our training samples. Lastly, the Colossal Clean Crawled Corpus (C4) [71], a large text-only dataset, is used for language model co-training, also making up 20% of our training samples.

**Diverse mixture of masking strategies.** As with 4M [65], the masking strategy is governed by Dirichlet distribution with parameter $\alpha$. This distribution influences the sampling of tokens from modalities: a lower $\alpha$ results in samples dominated by one modality, while a higher $\alpha$ leads a more balanced representation across all modalities. For both CC12M and COYO datasets, we implement multiple masking strategies to cater to specific training needs, and randomly sample from them for every sample in the batch:

- **All-to-all masking:** Involves four masking strategies with symmetric input and target $\alpha$ set to 0.01, 0.1, 1.0, and 10.0 respectively.

- **RGB-to-all masking:** Consists of only RGB tokens as input, with target $\alpha$ all set to 0.5.

- **Caption-biased masking:** Includes two strategies, heavily skewed towards either unmasked captions or T5-XXL embeddings as input. These masking strategies are particularly beneficial for tasks involving text-to-image generation

## F   Out-of-the-box Evaluation Details

Below, we provide further details on out-of-the-box evaluations we performed. Please also see fig. 14 for a qualitative comparison between our XL model and Unified-IO XL [61], as well as Unified-IO 2 XXL [60]. Furthermore, table 7 compares Unified-IO, Unified-IO 2, and our model's out-of-the-box capabilities on surface normal estimation, depth estimation, and semantic segmentation. As demonstrated, our model outperforms Unified-IO and Unified-IO 2 in all the mentioned tasks.

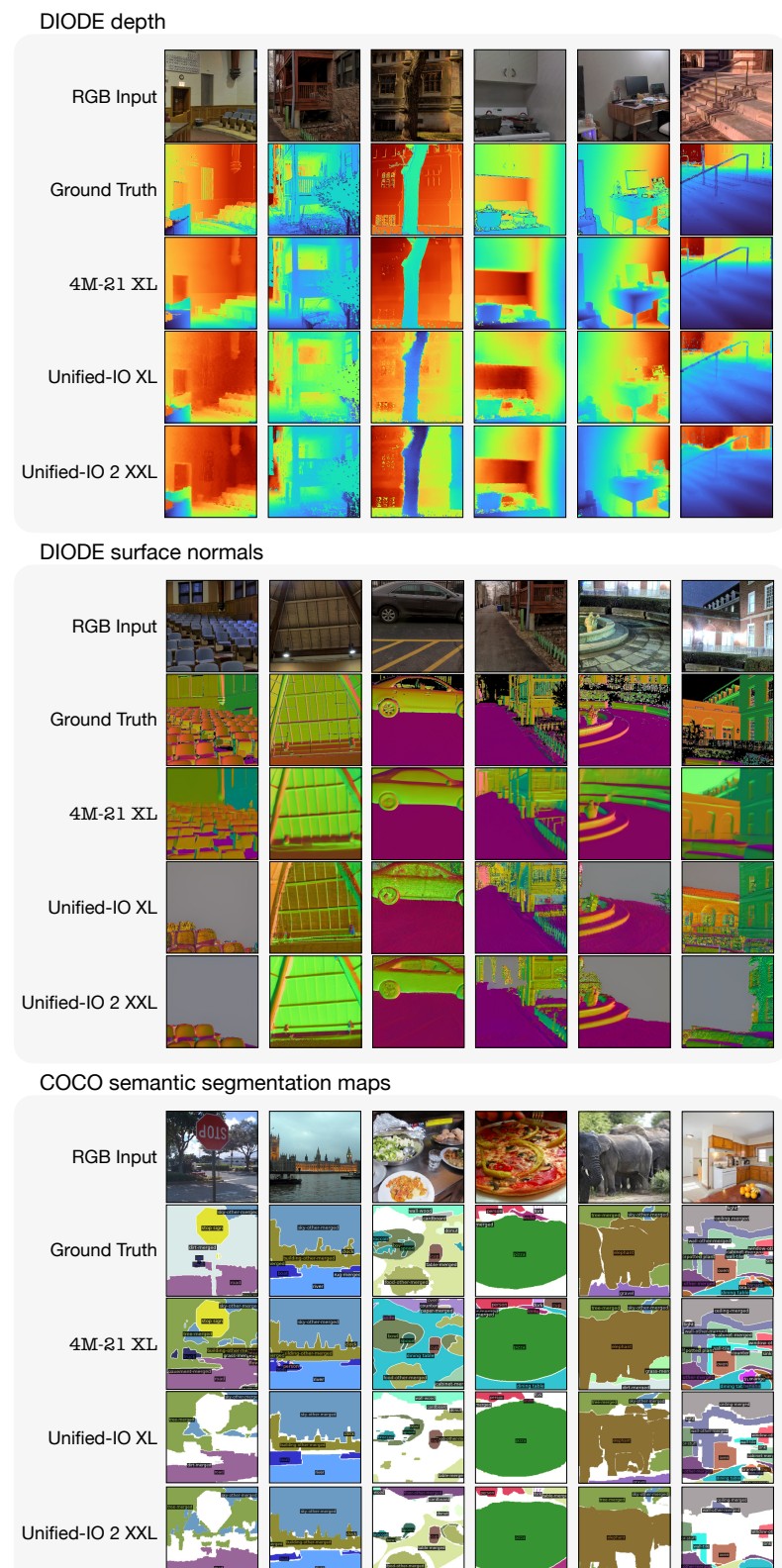

Figure 14: **Comparing** `4M-21` **XL, Unified-IO XL [61], and Unified-IO 2 XXL [60] out-of-the-box.** `4M-21` XL demonstrates strong generalization to inputs from different datasets and tasks out-of-the-box (zero shot), significantly improving over Unified-IO 1 and 2.

Table 6: **Pre-training settings.** Training configuration for 4M-21 used in the transfer experiments and generation results.

| Configuration | 4M-21 B | 4M-21 L | 4M-21 XL |
|---|---|---|---|
| Weight initialization | | 4M (COYO) | |
| Training length ($n$ tokens) | | 500B | |
| Warmup length ($n$ tokens) | | 10B | |
| Optimizer | | AdamW [59] | |
| Opt. momentum | | $\beta_1, \beta_2 = 0.9, 0.95$ | |
| Base learning rate [38] | 1e-4 | 1e-4 | 2e-5 |
| Batch size | | 8192 | |
| Weight decay | | 0.05 | |
| Gradient clipping | ✗ | ✗ | 3.0 |
| Learning rate schedule | | Cosine decay | |
| Feedforward activation | | SwiGLU [81] | |
| Input token budget | | 256 | |
| Target token budget | | 256 | |
| Input and target $\alpha$ | | Mixture (see Sec. E.2) | |
| Masking strategy | | Mixture (see Sec. E.2) | |
| Dataset | | Mixture (see Sec. E.2) | |
| Image resolution | | $224^2$ | |
| Augmentation | | None (Center Crop) | |
| Repeated sampling [32] | | 4 | |
| Data type | | bfloat16 [11] | |

Table 7: **Out-of-the-box capabilties.** Comparison between Unified-IO 2 and our model out-of-the-task capabilities across surface normal estimation, depth estimation, and semantic segmentation. We use the L1 score as the metric for surface normal and depth estimation, and mean IoU for semantic segmentation.

| Method | Normals ↓ | Depth ↓ | Sem. seg. ↑ |
|---|---|---|---|
| Unified-IO B [61] | 35.7 | 1.00 | 32.9 |
| Unified-IO L | 33.9 | 0.87 | 41.6 |
| Unified-IO XL | 31.0 | 0.82 | 44.3 |
| Unified-IO 2 L [60] | 37.1 | 0.96 | 38.9 |
| Unified-IO 2 XL | 34.8 | 0.86 | 39.7 |
| Unified-IO 2 XXL | 37.4 | 0.84 | 41.7 |
| 4M-21 B | 21.7 | 0.71 | 42.5 |
| 4M-21 L | 21.1 | 0.69 | 46.4 |
| 4M-21 XL | **20.8** | **0.68** | **48.1** |

## F.1 Surface normal and depth estimation on DIODE

We follow the evaluation setup in [65] and evaluate on DIODE validation set at $224 \times 224$ input resolution.

## F.2 Semantic and instance segmentation on COCO

We employ a similar approach as SAM [49] by querying our model on the bounding boxes to obtain the instances. To predict the instances, only the target bounding box is provided in the input final sequence, and the tokens are masked for our model to predict them.

## F.3 kNN retrieval on ImageNet-1K

We follow the evaluation setup from DINOv2 [68]and set $k = 20$ and temperature to 0.07.

## F.4    3D human pose prediction on 3DPW

We follow the evaluation implemented in the 4D-Humans [37] codebase, with the difference that we use $224 \times 224$ as input image resolution as opposed to $256 \times 256$.

# G    Transfer Evaluation Details

We provide the transfer settings in Tables 8, 9, 10. We also note that after an extensive hyper parameter search for the DINOv2-g baseline on NYUv2, using a ConvNeXt head, it achieved only 92.5 $\delta_1$ acc., which is lower than the reported 95.0 with frozen encoder and DPT head.

Table 8: **Image classification settings.** Configuration for intermediate fine-tuning on ImageNet-21K and fine-tuning on ImageNet-1K, the settings follow MultiMAE [7] and 4M [65].

| Configuration | ImageNet-21K | | | ImageNet-1K | | |
|---|---|---|---|---|---|---|
| | Base | Large | XL | Base | Large | XL |
| Fine-tuning epochs | | 20 | | 50 | 20 | 20 |
| Warmup epochs | | 2 | | | 2 | |
| Optimizer | | AdamW [59] | | | AdamW [59] | |
| Opt. momentum | | $\beta_1, \beta_2 = 0.9, 0.95$ | | | $\beta_1, \beta_2 = 0.9, 0.999$ | |
| Base learning rate [38] | 1e-4 | 1e-4 | 5e-5 | | 1e-4 | |
| Batch size | | 4096 | | 4096 | 4096 | 1024 |
| Weight decay | | 0.05 | | | 0.05 | |
| Learning rate schedule | | Cosine decay | | | Cosine decay | |
| Layer-wise lr decay [21] | 0.75 | 0.85 | 0.85 | 0.75 | 0.85 | 0.85 |
| Drop path [43] | 0.1 | 0.2 | 0.4 | 0.1 | 0.2 | 0.4 |
| Input resolution | | $224^2$ | | | $224^2$ | |
| Augmentation | | RandAug(9, 0.5) [22] | | | RandAug(9, 0.5) [22] | |
| Random resized crop | | (0.5, 1) | | | (0.08, 1) | |
| Label smoothing $\varepsilon$ | | 0.1 | | | 0.1 | |
| Mixup [103] | | 0.1 | | | 0.1 | |
| Cutmix [101] | | 1.0 | | | 1.0 | |

Table 9: **Semantic segmentation settings.** Configuration for semantic segmentation fine-tuning on ADE20K, the settings follow MultiMAE [7] and 4M [65].

| Configuration | ADE20K | | |
|---|---|---|---|
| | Base | Large | XL |
| Fine-tuning epochs | 64 | 64 | 128 |
| Warmup epochs | | 1 | |
| Optimizer | | AdamW [59] | |
| Opt. momentum | | $\beta_1, \beta_2 = 0.9, 0.999$ | |
| Learning rate | 2e-4 | 2e-4 | 3e-4 |
| Batch size | | 64 | |
| Weight decay | | 0.05 | |
| Learning rate schedule | | Cosine decay | |
| Layer-wise lr decay [21] | 0.75 | 0.85 | 0.95 |
| Drop path [43] | 0.1 | 0.2 | 0.3 |
| LoRA [42] rank / scale | ✗ | ✗ | 64 / 1.0 |
| Input resolution | | $512^2$ | |
| Augmentation | | Large-scale jitter (LSJ) [34] | |
| Color jitter | | ✓ | |

Table 10: **Depth estimation settings.** Configuration for depth estimation fine-tuning on NYUv2, the settings follow MultiMAE [7] and 4M [65].

| | NYUv2 | | |
| Configuration | Base | Large | XL |
| --- | --- | --- | --- |
| Fine-tuning epochs | | 1000 | |
| Warmup epochs | | 100 | |
| Optimizer | | AdamW [59] | |
| Opt. momentum | | $\beta_1, \beta_2 = 0.9, 0.999$ | |
| Learning rate | 1e-4 | 1e-4 | 5e-5 |
| Batch size | 128 | 128 | 16 |
| Weight decay | | 1e-4 | |
| Learning rate schedule | | Cosine decay | |
| Layer-wise lr decay [21] | 0.75 | 0.85 | 0.9 |
| Drop path [43] | 0.1 | 0.2 | 0.0 |
| LoRA [42] rank / scale | ✗ | ✗ | 8 / 1.0 |
| Input resolution | | $256^2$ | |
| Random crop | | ✓ | |
| Color jitter | | ✓ | |

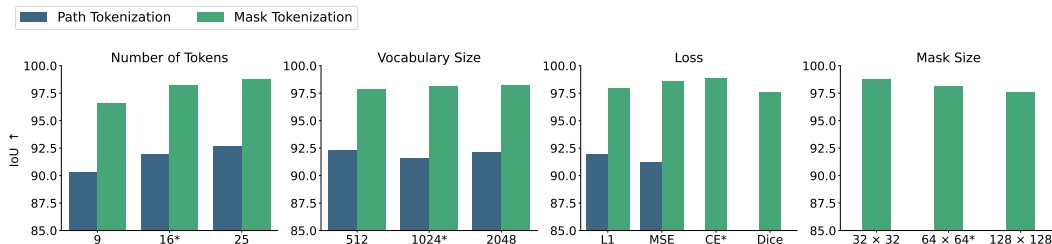

Figure 15: **Ablating tokenization choices:** We ablate the impact of different tokenization choices. Performance is reported as reconstruction IoU on CC12M validation set. * shows the mask tokenization configuration we used in the final tokenizer. See appendix H for details.

# H   Investigating Different Tokenization Schemes

As we develop several tokenization strategies for each modality, ablating their performance against all possible design choices would be prohibitively expensive. Thus, we focus on one modality, namely SAM instances, and provide a more detailed look into the impact of different tokenization strategies. We study two approaches for SAM instances: *path tokenization* and *mask tokenization*.

**Path tokenization**: We represent each instance in the image as a list of polygon coordinates. Then we tokenize these coordinates using a Bottleneck MLP-based VQ-VAE tokenizer. To achieve a fixed-size input, the polygons are either simplified or extended to have the same number of corner points. We found that fixing the maximum number of corners to 128 results in a minimal change in the overall polygon shape, thus we use this value for all the path tokenization ablations.

**Mask tokenization**: In this scheme, we first convert each instance to a binary masks and resize them to a fixed mask size. Then, we tokenize them using a ViT-based VQ-VAE tokenizer, similar to the way we tokenize image-like and feature map modalities.

**Ablations:** We investigated L1 and MSE losses for both tokenization schemes, and additionally cross-entropy and Dice loss for the mask tokenization. We also investigated the effects of the total number of tokens, token vocabulary size, and mask size. To compare the performance of the resulting tokenizers, we use the IoU between the pseudo-labeled and reconstructed instances as our metric. fig. 15 illustrates the results of different ablated configurations. For each configuration, the remaining unspecified parameters are by default set to 16 for the number of tokens, 1024 for the vocabulary size, $L1$ for the loss, and $64 \times 64$ for the mask size. The ablations show that using mask tokenization with 16 tokens, 1024 vocabulary size, and $64 \times 64$ mask size performs well and sets a good balance between reconstruction quality and total sequence length.

In all ablations, the tokenizers are trained for 24 epochs starting with 5 warmup epochs using the AdamW [59] optimizer with $\beta_1, \beta_2 = 0.9, 0.999$ and a batch size of 128. For all the experiments except the Dice loss, a learning rate of 1e-5 is used. Since using this learning rate for the Dice loss experiment resulted in instabilities, we reduced its learning rate to 1e-6. As demonstrated in fig. 16, increasing the number of tokens results in better reconstruction quality both for the mask tokenizer and the path tokenizer. Compared to L1 loss, the cross-entropy loss training obtains reconstructions with smoother edges and better coverage.

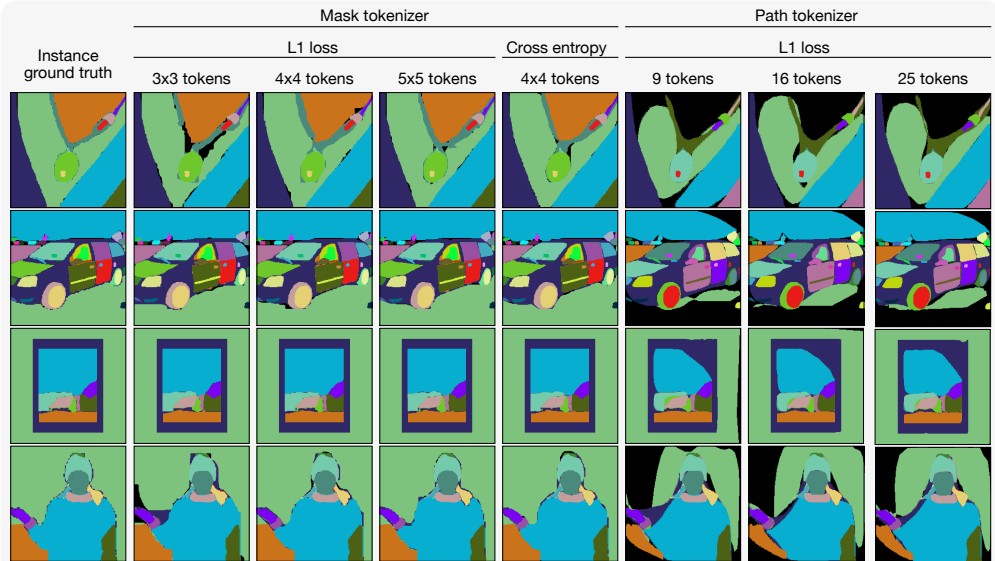

Figure 16: **Different tokenization schemes for SAM instances.** We compare different tokenization schemes to tokenize SAM instances for pre-training. Please see Sec. H for details.

# I Additional Discussions and Results

## I.1 Encoder-decoder vs decoder-only architectures

In our work, we decided to use an encoder-decoder architecture to achieve any-to-any multimodal capabilities, while other approaches [74, 87, 92, 10] use decoder-only architectures in multimodal settings. Both architectural choices are valid depending on specific use cases and priorities, and future work could explore these trade-offs more systematically. Below, we explain our choice of an encoder-decoder architecture and discuss how a similar model might look in a decoder-only setting.

Following 4M, we use an encoder-decoder architecture as it is directly compatible with masking approaches (e.g., T5 [71], MAE [40]), which is at the core of our method and enables any-to-any capabilities with a single training objective. In addition, after training, the encoder can be extracted and used as a ViT or a multimodal encoder. Notably, the bidirectional self-attention in the encoder has been shown to have slight benefits over causal attention for representation learning & transfer tasks [71, 30].

If we were to design a 4M-like model using a decoder-only architecture, there are two main approaches to consider:

The first is a causal decoder (i.e. next token prediction without span [71] or MAE [40] masking). This approach is similar to multimodal LLMs operating on interleaved data [74, 87, 92, 10] and is the easiest to unify with LLMs. However, the 4M masking strategy would not be directly compatible with this approach. A naive strategy would be to keep everything unmasked and concatenate one modality after another, but this leads to much larger sequence lengths per example and redundancy between modalities. As many of the capabilities shown in our paper rely on cross-modal masking (e.g. the any-to-any capabilities and the ability to predict outputs from partial modalities), it is unclear

Table 11: **Quantitative evaluation of image generation.** We compare our model's text-to-image generation performance against 4M, Stable Diffusion 2.1-base, as well as a controlled text-to-image specialist variant of 4M (T2I-B) trained with full text and masked RGB token inputs, conceptually similar to Muse. We compute FID and CLIP-L/14 scores on COCO validation set after resizing the generations to 256x256. All runs use guidance scale 3.0. 4M-21 consistently outperforms 4M by a good margin. 4M-21 can match the performance of the controlled Muse baseline, but not that of dedicated text-to-image models like SD-2.1, which are trained on orders of magnitude more data and compute.

| Model | Res. | Dataset | A100 hours | Text encoder | COCO val (30k) | |
| --- | --- | --- | --- | --- | --- | --- |
| | | | | | FID ↓ | CLIP score ↑ |
| 4M-7 B | 224 | CC12M | 2.3k | - | 37.5 | 21.4 |
| 4M-21 B | 224 | CC12M & COYO & C4 | 3.1k | - | **34.5** | **22.1** |
| 4M-7 L | 224 | CC12M | 9.2k | - | 30.1 | 23.1 |
| 4M-21 L | 224 | CC12M & COYO & C4 | 12.3k | - | **27.2** | **23.7** |
| 4M-7 XL | 224 | CC12M | 24.5k | - | 27.0 | 23.7 |
| 4M-21 XL | 224 | CC12M & COYO & C4 | 33.8k | - | **26.0** | **24.3** |
| T2I-B "Muse" | 224 | CC12M | 1.6k | - | 39.8 | 20.3 |
| SD-2.1-base | 512 | Curated LAION-5B | > 200k | CLIP ViT-L/14 | 10.1 | 25.5 |

whether those would be achievable with a causal decoder and what engineering challenges this would involve.

The second approach is using a prefix LM-like decoder (see Fig. 4 of T5 [71]), where unmasked inputs (i.e., encoder inputs in the 4M formulation) and masked inputs/targets (i.e., decoder inputs in the 4M formulation) are concatenated. The entire sequence is then given to a single decoder LLM. This approach allows preserving the masking strategy and training objective within a decoder-only architecture, but has seen less adoption than encoder-decoder approaches in the masking literature. However, it is more amenable to multi-turn or temporal inputs, as multiple sequences of unmasked inputs and masked targets can be concatenated one after the other.

In summary, we use an encoder-decoder architecture as it provides a straightforward way to achieve any-to-any capabilities through masking, and allows for downstream reuse of the trained encoder as a singlemodal or multimodal backbone. While decoder-only approaches could potentially be adapted for similar purposes, we are unaware of work demonstrating this at the same scale (in terms of number of modalities) and we believe it to be a very impactful and exciting research direction.

## I.2 Quantitative evaluation of image generation

In table I.2, we quantize our model's conditional generation capabilities by performing caption-to-image generation on COCO. We compare against 4M-7 [65] across all model sizes, Stable Diffusion 2.1 [76], and a controlled text-to-image specialist baseline. The controlled text-to-image baseline (T2I-B), conceptually similar to Muse [14], uses the same architecture and RGB tokenizer as our model and 4M-7, and was trained for a total of 300B tokens on CC12M [16]. We test for image fidelity using FID and image-text alignment using CLIP score, computed using 30'000 validation set images, and resizing all images to 256x256. We used guidance scale 3.0 for all experiments.

Our models are able to consistently outperform 4M-7 across model sizes on COCO, both in terms of FID and CLIP score. While there is still a seizable gap between dedicated text-to-image models like Stable Diffusion 2.1 and our models on out-of-distribution data, we note that these models are usually trained on orders of magnitude more data and compute. Our T2I-B baseline attempts to control for factors such as the tokenizer that can have a significant influence on FID, and we see that 4M-7 B performs similar to the specialist T2I-B. Optimizing for image generation quality was not the focus of our work, but considering the scaling trends of token-based masked (e.g. Muse [14], MAGE [55]) and auto-regressive models (e.g. Parti [96]), we expect significant improvement with larger model sizes. Furthermore, we expect that recent advances in RGB tokenization (e.g. MAGVIT-v2 [98], FSQ [63]) will translate to significant gains in FID for large enough models.

## I.3 Quantitative evaluation of cross-modal retrieval

We provide additional quantitative results for cross-modal retrieval in table I.3. Our model has notable performance for different retrieval tasks (RGB-Text, RGB-Depth, RGB-Semantic) while being only trained on global embeddings extracted from the RGB images.

Table 12: **Cross-modal retrieval quantitative results.** We report the performance on ImageNet1K and Flickr30k benchmarks (no fine-tuning). For all evaluations, Top-1 / Top-5 accuracies are reported. For cross-modal retrieval on Flickr30k, we perform both RGB→Text (i.e. R→T) and vice versa. For ImageNet, we use the pseudo-labeled depth (D) and semantic segmentation (S) since the dataset does not come with labels. Retrieval is performed using the method described in section 3.2. Cross ($\times$) means the model is not capable of performing that task due to not being a multimodal model. Dashed (-) means no official result was reported. We make the following observations: **1)** Our model successfully matches the performance of DINOv2 and ImageBind for RGB→RGB (R→R) retrieval, which is their primary domain, despite being a multitask model trained to do several other tasks as well. This indicates that our model distilled their embeddings well via masked modeling objective. **2)** On top of that, our model can perform cross-modal retrieval with notable performance across different modalities as shown below. Despite ImageBind being technically capable at cross-modal retrieval like R→D, no official result was reported in the paper. We tried to produce that number which led to poor results by ImageBind, so we refrain from reporting.

| Model | ImageNet1K kNN | | | Flickr30k | |
|---|---|---|---|---|---|
| | R $\rightarrow$ S | R $\rightarrow$ D | R $\rightarrow$ R | R $\rightarrow$ T | T $\rightarrow$ R |
| 4M-21 | 56.89/79.32 | 66.82/85.95 | 78.3/92.4 | 58.0/82.5 | 42.4/71.6 |
| ImageBind | - | - | 81.1/94.4 | - | - |
| DINOv2 | $\times$ | $\times$ | 82.1/93.9 | $\times$ | $\times$ |
| CLIP | $\times$ | $\times$ | 79.59/94.0 | 88.0/98.7 | 68.7/90.6 |

# J Broader Impact

## J.1 Computational costs

All models were trained on Nvidia A100 GPUs. The 4M-21 B model was trained for 2 days using 64 A100s. The 4M-21 L model was trained for 4 days using 128 A100s. The largest 4M-21 XL model required 11 days using 128 A100s. Fine-tuning and transfer learning experiments for each model used approximately 20% additional compute compared to its pre-training. Training the various tokenizers (RGB, depth, normals, CLIP, DINOv2, ImageBind, semantic segmentation, SAM edges, and Canny edge detection, SAM instances, and 3D human poses) required roughly 5 days using 8 A100s each, totaling approximately 60 A100-days. In total, the primary experiments reported in the paper used approximately 120'000 A100-hours, not including additional preliminary experiments and ablations. We estimate the total compute for the full research project, including preliminary and unreported experiments, to be 150'000 A100-hours.

## J.2 Social impact

We are open sourcing our code and models to support researchers with the democratization of the tools and to enable transparent inspection and safeguarding. 4M-21 models are trained on publicly available datasets with some curation, e.g. people's names are redacted in CC12M [16]. However, this process is still noisy, hence we advise caution when using the models for generation.

