# OpenReview forum: "4M-21: An Any-to-Any Vision Model for Tens of Tasks and Modalities"
_NeurIPS.cc/2024/Conference — NeurIPS 2024 poster_

### Official Review · Reviewer_66Uu · 2024-07-04

**Soundness:** 3
**Presentation:** 3
**Contribution:** 3
**Rating:** 8
**Confidence:** 5

**Summary:**

The paper presents an advanced vision model capable of handling a wide range of tasks and modalities, demonstrating the potential to train a single model on tens of diverse modalities without a loss in performance compared to specialized models. Specifically, the model is trained on a multitude of modalities, including RGB images, depth, semantic segmentation, CLIP features, surface normals, and more, enabling it to perform various tasks such as image generation, retrieval, and understanding.

**Strengths:**

+ The presentation is clear and detailed.
+ The performance of the proposed model is good.
+ The workload of this paper is impressive.

**Weaknesses:**

1. Do the authors try to measure the quality (e.g. FID ) of generated images (based on caption input) and can the proposed method surpass some existing diffusion-based models?

2. The authors should illustrate the detailed configurations of B, X, and XL models, for example, the number of layers and the width of channel dimensions.

**Questions:**

see weakness

**Limitations:**

see weakness

---

> ### Author Rebuttal · Authors · 2024-08-07
>
> We thank reviewer 66Uu for the positive feedback. We address the main concerns and questions below:
>
> > Quality of generated images
> >
>
> Please see the `PDF` for detailed caption-conditioned generation metrics on COCO, as well as Section 2 of the common response for a discussion.
>
> > Detailed configurations of B, L, and XL models
> >
>
> Thank you for pointing this out. We've listed the model configurations in the table below. These configurations are derived from 4M and closely resemble those of T5 and UnifiedIO to ensure comparability with other encoder-decoder models in the literature.
>
> Note that we use SwiGLU [15] in the feedforward, which employs three weight matrices instead of the typical two. To maintain equivalent parameter and operation counts, we've adjusted the feedforward dimension by scaling it by 2/3 (4 x 2/3 instead of 4x).
>
> | Model    | Encoder Blocks | Decoder Blocks | Model Dim | Feedforward Dim | Num Heads | Total Params |
> |----------|----------------|----------------|-----------|-----------------|-----------|--------------|
> | Ours B  | 12             | 12             | 768       | 2048            | 12        | 198M         |
> | Ours L  | 24             | 24             | 1024      | 2730            | 16        | 705M         |
> | Ours XL | 24             | 24             | 2048      | 5461            | 32        | 2818M        |

---

### Official Review · Reviewer_PHF3 · 2024-07-04

**Soundness:** 3
**Presentation:** 3
**Contribution:** 2
**Rating:** 6
**Confidence:** 4

**Summary:**

The paper addresses the limitations of current multimodal and multitask foundation models, such as 4M and UnifiedIO, which are constrained by the limited number of modalities and tasks they can handle. The authors present a model trained on a wide variety of modalities and tasks using large-scale multimodal datasets and text corpora. This includes training on images, text, semantic and geometric modalities, feature maps from state-of-the-art models, and new modalities like image metadata and color palettes. A key technique used is discrete tokenization of various data types. The new model can handle at least three times more tasks and modalities than existing models without losing performance. This approach also enhances fine-grained and controllable multimodal generation and explores the unification of diverse models into a single one.

**Strengths:**

1. The paper has a good starting point, noting that the network structures across various AI fields are converging (mostly to transformers).

2. This paper involves a significant amount of engineering work. Organizing large amounts of data and conducting large-scale training is not easy.

3. The experiments and performance in the paper are quite good.

**Weaknesses:**

1. The writing of the paper has room for improvement. Not all reviewers have read the 4M paper. The paper does not detail the training framework, such as how to control different tasks and how to conduct multimodal masked training. These aspects are not simply described and are only mentioned in section 2 as following the 4M paper. Including a simple diagram would significantly improve clarity.

2. Some claims are somewhat inaccurate. There are works that use very simple methods to unify modeling for multiple tasks, maybe not as many as 10. It is necessary to discuss the difference and challenges between completing 5 tasks and 10+ tasks, as some tasks can actually be categorized as a single task.

**Questions:**

1. The paper uses an encoder-decoder paradigm, whereas we know that the commonly used LLMs nowadays typically employ a decoder-only structure or simply a multi-layer transformer structure. Why does the paper adopt the encoder-decoder paradigm? The decoder-only approach[1][2] seems simpler and better suited for unifying with the LLM structure.
The authors are advised to include some discussion on encoder-decoder and decoder-only approaches, referencing papers such as the following.

[1] GiT: Towards Generalist Vision Transformer through Universal Language Interface. (ECCV 2024)

[2] Fuyu-8B: A Multimodal Architecture for AI Agents. (Blog)
The two articles mentioned above also tokenize everything within a simple unified framework.

2. Do different feature maps count as different modalities? I thought different modalities referred to the basic levels such as image, language, and speech, etc. The formulation in the paper is somewhat strange, and I hope the authors can explain this a bit. After all, these features are generated by cost-intensive multi-modal encoders, which differs from the lightweight multimodal tokenizer proposed in the paper. I also hope the authors can discuss this in comparison with the approaches of GiT[1] and Fuyu-8B[2], which use light-weight yet simple tokenizers on original image and language.

3. Regarding the challenge of negative transfer, you can also refer to the GiT paper. It demonstrated that joint training on five tasks has better performance than single-task training. (Considering that this paper is also recent, a direct comparison may not be necessary, but it is still recommended to discuss it.) :)

Overall, I am inclined to accept this paper, but I still hope the authors can address my concerns. If these issues are addressed well, I will accordingly raise my score. :)

**Limitations:**

There are no comments regarding the limitations.

---

> ### Author Rebuttal · Authors · 2024-08-07
>
> We thank reviewer PHF3 for the positive feedback. We address the main concerns and questions in the following response:
>
> > Encoder-Decoder vs Decoder-only.
>
> Thank you for your question. **Both encoder-decoder and decoder-only architectures are valid design choices, depending on the specific use cases and priorities, and this is somewhat that can be explored more solidly in future work.** We explain the reasons for our choice of an encoder-decoder architecture below and discuss how a similar model might look like in a decoder-only setting.
>
> Following 4M, we use an encoder-decoder architecture as it is **directly compatible with masking approaches (e.g., T5, MAE), which is at the core of our method and enables any-to-any capabilities with a single training objective**. In addition, **after training, the encoder can be extracted and used as a ViT or a multimodal encoder**.  Notably, the bidirectional self-attention in the encoder has been shown to have slight benefits over causal attention for representation learning & transfer tasks (e.g., Table 3d of AIM [9], Table 2 of T5 [10]).
>
> If we were to design a 4M-like model using a decoder-only architecture, there are two main approaches to consider:
>
> The first is a causal decoder (i.e. next token prediction without span or MAE masking). This approach is similar to multimodal LLMs operating on interleaved data (e.g. Gato [11], Fuyu [12], GiT [13], Chameleon [19]) and is the easiest to unify with LLMs. However, the 4M masking strategy would not be directly compatible with this approach. A naive strategy would be to keep everything unmasked and concatenate one modality after another, but this leads to much larger sequence lengths per example and redundancy between modalities. **As many of the capabilities shown in our paper rely on cross-modal masking (e.g. the any-to-any capabilities and the ability to predict outputs from partial modalities), it is unclear whether those would be achievable with a causal decoder and what engineering challenges this would involve.**
>
> The second approach is using a prefix LM-like decoder (see Fig. 4 of T5 paper [10]), where unmasked inputs (i.e., encoder inputs in the 4M formulation) and masked inputs/targets (i.e., decoder inputs in the 4M formulation) are concatenated. The entire sequence is then given to a single decoder LLM. **This approach allows preserving the masking strategy and training objective within a decoder-only architecture, but has seen less adoption than encoder-decoder approaches in the masking literature.** However, it is more amenable to multi-turn or temporal inputs, as multiple sequences of unmasked inputs and masked targets can be concatenated one after the other.
>
> In summary, **we use an encoder-decoder architecture as it provides a straightforward way to achieve any-to-any capabilities through masking, and allows for downstream reuse of the trained encoder**. While decoder-only approaches could potentially be adapted for similar purposes, we are unaware of work demonstrating this at the same scale (in terms of number of modalities) and we believe it to be a very impactful and exciting research direction.
>
> As suggested, we will include additional references to decoder-only multimodal LLMs as well as parts of this discussion in our camera-ready version.
>
> > Do different feature maps count as different modalities? Comparison with approaches of GiT and Fuyu-8B.
>
> This highlights an important point; thanks for the question. Three clarifications:
>
> 1) our primary goal in this paper is ‘modeling’ work, e.g. **demonstrating the possibility of having one model that effectively does any-to-any prediction across many diverse modalities**. Once that is established, the community can adopt the model on their own modalities that are justified based on the use case. **Our goal is not committing a specific dictionary of modalities or ruling out others, but using a set that provides diversity and is relatively large to enable evaluation.**
>
> 2) **‘what qualifies as a modality’ is an interesting nontrivial question.** Many modalities, e.g., depth/3D, heat, etc., can be sensed using sensors, but also, they can be effectively inferred out of RGB images with a powerful task-specific prediction model. Being inferable with a task-specific neural network does not question if depth/3D qualifies as a modality of information. **In general, in theory, any specific mode of information about an underlying scene qualifies as a modality.** Which modalities are ‘useful’ is a different deeper question, often application specific, and out of our scope as the paper’s focus is making progress on the model.
>
> 3) adopting neural network feature maps as a modality is useful as, they can be **used for chained conditioning [18], and also they provide a way for distilling popular single networks (e.g. DINO) into one multitask network**.
>
> Similar to GiT, we employ tokenization of several modalities via language, e.g. bounding boxes, metadata and color palette. However, this strategy is limited for tokenizing dense representations such as depth, normals, edges, instance segmentation masks, or feature maps. **Thus, we employ different tokenization schemes to handle such modalities as depicted in Figure 3.** We will add a discussion on this to the paper.
>
> > Challenge of negative transfer.
>
> Thanks for the pointer, we will add a discussion. While we didn’t observe negative transfer when more modalities are included, investigating this direction further could be worthwhile.
>
> > Detail the training framework
>
> Thanks. We will add a diagram to camera-ready to improve the clarity as suggested.

---

> > ### Comment · Reviewer_PHF3 · 2024-08-13
> >
> > Thank you for the detailed response. I hope the author will include the contents of the rebuttal in the camera-ready version, including the discussion on Decoder-only models and feature maps as a modality. This will provide more insights to those who read the paper. I will accordingly raise my score.

---

> > > ### Author Response · Authors · 2024-08-14
> > > **Thank you!**
> > >
> > > We thank the reviewer for the valuable suggestions and positive feedback. We will include the contents of the rebuttal in the camera ready.

---

### Official Review · Reviewer_b2ju · 2024-07-15

**Soundness:** 2
**Presentation:** 4
**Contribution:** 2
**Rating:** 5
**Confidence:** 4

**Summary:**

The authors present a new vision model that can generate tokens in all directions that represents multiple modalities like RGB images, depth map, segmentation maps, color palette, DINOv2 features etc. given a conditioning on a subspace of the modalities.
They develop a new way to tokenize certain modalities and scale both model’s parameters and dataset’s size.

**Strengths:**

- Good performances on a lot of downstream tasks. The authors evaluate the model on a wide variety of tasks and show promising results that are either on-par or above very strong baselines.
- Showing scaling to bigger models isn’t easy
- Multimodal retrieval is interesting to see and could be leveraged for a lot of new interesting tasks.

**Weaknesses:**

- The added value over 4M is mostly due to engineering.
- No multimodal retrieval nor conditional generation metrics. The actors emphasize a lot the generative/multimodal part of the model but the only quantitative experiments are shown in Table 3. Qualitative figures are interesting but should be supporting quantitative results. The conditional generation evaluation needs to be compared to other models like ControlNet [1] and you should at least provide precision/recall@k on some modalities for the multimodal retrieval (like comparing DINOv2/ImageBind with one modality against your model with one modality and your model with multiple ones).
- The tokenization might reduce the burden of finding good losses’ weight, but tokenizers trained on a specific domain (like CC12M) might not generalize well. Moreover, tokenizers might not capture high frequency details that are very important like text for OCR, good quality faces, or smaller details. In [2], the authors speak about the fact that “quality limitations of the VQ image reconstruction method inherently transfer to quality limitations on images”. They also focus on the different domains that are not well captured by the tokenizers. The current paper only includes one ablation on the tokenizer, and I’m skeptical of the tokenizers’ performances on out-of-distribution images.

[1] Adding Conditional Control to Text-to-Image Diffusion Models, Zhang, Rao and Agrawala
[2] Make-A-Scene: Scene-Based Text-to-Image Generation with Human Priors, Gafni et al.

**Questions:**

- What are the performances of the trained tokenizers? Given that your model will be somewhat bounded by the tokenizers’ performances, could you provide a table including the performances of all tokenizers versus the original model and yours?
- How did you chose each vocabulary size?
- The metadata picked by the authors seem arbitrary, could you explain your decision process?
l.223 I don’t understand why you treat RGB tokens and pixels differently: ‘we leverage RGB patch embeddings learned during the pre-training, as RGB pixel inputs are used alongside the tokenized modalities’. Could you expand on this please?

The paper is interesting nonetheless but could be way, way stronger with quantitative evaluations on the new claimed capabilities.

**Limitations:**

The authors talk about all the limitationsof their work.

---

> ### Author Rebuttal · Authors · 2024-08-07
>
> We thank reviewer b2ju for the constructive feedback. We address the main concerns and questions in the following response:
>
> > What are the performances of the trained tokenizers?
>
> Please see the following items as the response: 1) Figure 1 in the rebuttal `PDF` for a visual comparison of multimodal tokenizer reconstructions (*resolution 224x224*), 2) the “tokenizer bound” reported in the main paper's Table 1 that represents the performance upper bound — it clearly shows that the prediction errors of all models are notably worse than the tokenization error indicating that the **lossiness of discrete tokenization is not the bottleneck**.
>
> > Generalization of tokenizers
>
> We train tokenizers on CC12M to match our training data distribution, and find them to perform well within that distribution (see above answer). For more niche domains not captured in web-scale image distributions (medical, satellite, etc.), we recommend training domain-specific tokenizers. That said, we note that **tokenizers trained on web-scale data perform significantly better in areas like human face and text reconstruction** compared to popular tokenizers like VQ-GAN [1] that were trained on ImageNet-1k; a finding supported by MAGVIT-v2 [2].
>
> > Capturing high-frequency details
>
> This is challenging for discrete tokenizers, but can be addressed by increasing token density and performing an additional token super-resolution step, similar to 4M and Muse. Furthermore, recent advances in VQ-free tokenization [2,3] show strong scaling trends in reconstruction and generation performance relative to the vocabulary size.
>
> We would like to stress that **discrete tokenization is one of the key design decisions that allows us to scale a unified model to such a large and diverse set of modalities**, without multi-task loss-balancing, task-specific heads, or encountering training instabilities. We expect the above-mentioned advances in discrete tokenization to show direct benefits in multimodal generation and out-of-the-box performance.
>
> > Multimodal retrieval numbers
>
> Please refer to the `PDF` Tab. 2 for additional evaluations on cross-modal retrievals. **Our model has notable performance for different retrieval tasks (RGB-Text, RGB-Depth, RGB-Semantic) while being only trained on global embeddings extracted from the RGB images.**
>
> Regarding DINOv2, it is trained only on RGB images and is not a multimodal model, thus capable of only RGB-RGB retrieval. We also compared our model’s RGB retrieval performance with DINOv2/ImageBind using ImageNet kNN classification, (Tab. 1 of the main paper, also included in the `PDF`). The results in the table show that **our model’s RGB retrieval capability successfully matches DINOv2 and ImageBind, and on top of that, it can perform multimodal retrieval which DINOv2 and other similar models cannot.**
>
> > Conditional generation numbers
>
> Please see the `PDF` for detailed caption-conditioned generation metrics on COCO, as well as Section 2 of the common response for a discussion.
>
> > How did you choose each vocabulary size?
>
> We follow the best practices from community and tune the vocabulary size to obtain low reconstruction error while keeping the vocabulary size as small as possible. Please see Fig. 15 and Appendix J for an ablation.
>
> > Metadata decision process
>
> Our choice of which metadata to include was guided by Omnidata [4] and SDXL [5]. Omnidata extracts several types of metadata from a multimodal dataset to give researchers a high-level set of controls to steer the data distribution towards high/low walkability, occlusion, etc. Our goal was to expose similar parameters, but for a multimodal generative model. **We believe that such capabilities provide a path to answer questions on what the ideal multimodal data distributions are for different downstream tasks.**
> Furthermore, we looked towards SDXL and more broadly ControlNet [6] as inspiration for metadata that can be used for image generation, e.g. original image size, colorfulness, etc. While many types of metadata can be included as text in a prompt, we observe strong condition adherence with metadata. In summary, our aim was to provide a broad set of control knobs by extracting various parameters from RGB images and the different pseudo labels. **We note that this set can be easily extended towards scores like aesthetics, NSFW, watermark, etc.**
>
> > Treating RGB tokens and pixels differently
>
> Thank you, this is an important design choice in current multimodal models.
>
> Discrete tokens enable iterative sampling, making them useful for generative tasks, which is why most token-based generative methods (e.g., Parti [7], MaskGiT [8]) use them. However, discretization leads to information loss, which is not ideal for visual perception tasks.
> On the other hand, using RGB pixels as input is more suitable for visual perception tasks. By avoiding the discrete bottleneck, there is no information loss during the tokenization step, and the projection layer can be more lightweight (in 4M's case, it is just a simple linear projection).
> Given these tradeoffs, we follow 4M by training on both and treating them as separate modalities, with RGB pixels as an input-only modality. **This allows us to choose the most appropriate representation for a given task - discrete RGB for generation or RGB pixels for perception.**
>
> > The added value is mostly due to engineering.
>
> We find it hard to concretely respond to this comment as a weakness since, first, many undeniably key contributions in the community are basically “engineering”, and second, what would be “engineering” is too vague and broad. **Engineering doesn’t mean unimpactful.** All reviewers positively recognized that this paper required a significant amount of exploration and work, e.g., in terms of scaling, and the workload was viewed as a strength of the paper. Besides those, the paper made several contributions and introduced new capabilities across key axes, as summarized in L52-L70 in the main paper.

---

> > ### Comment · Reviewer_b2ju · 2024-08-11
> >
> > Thanks for your detailed answer.
> > I really appreciate the tables 1 and 2 in the rebuttal pdf and the references about tokenizers' quality.
> >
> > A CLIP baseline in the Table 2 for image and text retrieval would perfect it.
> >
> > **Treating RGB tokens and pixels differently**
> > Adding a small discussion about this in the camera ready would help readers understand your thought process and why it is done like this.
> > A small table comparing results with RGB vs tokenizer as input would also help readers chose the best way for their own task when using your model.
> >
> > **The added value is mostly due to engineering.**
> > I apologize for the confusing term I used. I was pointing out that your comparison uses 4M, but you also utilized more compute and data, which I wrongfully referred to as engineering. The new capabilities you've added are impressive, particularly with the quantitative results to compare against the pseudo-labelers. However, it's unclear whether the improvements on previous capabilities (wrt. 4M) are due to the additional data or because increasing the number of modalities leads to compounding gains on the other ones.
> >
> > I think that the paper now pass the bar for acceptance as long as you include the Table 1 and Table 2 from rebuttal.
> > Thanks again for resolving my issues and I'll update my score.

---

> > > ### Author Response · Authors · 2024-08-14
> > > **Thank you!**
> > >
> > > We thank the reviewer for the valuable suggestions and positive feedback. We will include the discussions and results in the camera ready.

---

### Author Rebuttal · Authors · 2024-08-07

## **Response to all reviewers**

We thank the reviewers for their insightful comments and constructive feedback. We are pleased that they commended our performance with remarks such as: **“good performances on a lot of downstream tasks”** (b2ju), **“The experiments and performance in the paper are quite good”** (PHF3), and “**The performance of the proposed model is good”** (66Uu).

The reviewers also recognized our scaling efforts: **“Showing scaling to bigger models isn’t easy”** (b2ju), **“This paper involves a significant amount of engineering work. Organizing large amounts of data and conducting large-scale training is not easy”** (PHF3), and **“The workload of this paper is impressive”** (66Uu).

Additionally, we are glad that the reviewers found our results promising that **“could be leveraged for a lot of new interesting tasks”** (b2ju), and praised our presentation as **“clear and detailed”.** (66Uu)

### 1. Additional results overview

We address the reviewers’ remaining questions and concerns in the individual responses and rebuttal `PDF` . We discuss the questions on the quality of generated images below. We also provide a list of additional results to address reviewer questions:

- b2ju, 66Uu: Conditional generation numbers (`PDF` Tab. 1)
- b2ju: Tokenizer performance (`PDF` Fig. 1)
- b2ju: Multimodal retrieval numbers (`PDF` Tab. 2)

### 2. Common questions

> b2ju, 66Uu: Quality of generated images
>

In Tab. 1 of the rebuttal `PDF`, we quantize our model's conditional generation capabilities by performing caption-to-image generation on COCO. We compare against 4M [18] across all model sizes, Stable Diffusion 2.1, and a controlled text-to-image specialist baseline. The controlled text-to-image baseline (T2I-B), conceptually similar to Muse [16], uses the same architecture and RGB tokenizer as our model and 4M, and was trained for a total of 300B tokens on CC12M. We test for image fidelity using FID and image-text alignment using CLIP score, computed using 30'000 validation set images, and resizing all images to 256x256. We used guidance scale 3.0 for all experiments.

**Our models are able to consistently outperform 4M across model sizes on COCO, both in terms of FID and CLIP score**.

While there is still a seizable gap between dedicated text-to-image models like SD2.1 and our models on out-of-distribution data, we note that **these models are usually trained on orders of magnitude more data and compute**. Our T2I-B baseline attempts to control for factors such as the tokenizer that can have a significant influence on FID, and we see that Ours-B performs similar to the specialist T2I-B.

Optimizing for image generation quality was not the focus of our work, but considering the scaling trends of token-based masked (e.g. Muse [16], MAGE [17]) and auto-regressive models (e.g. Parti [7]), **we expect significant improvement with larger model sizes**. Furthermore, we **expect that recent advances in RGB tokenization** (e.g. MAGVIT-v2 [2], FSQ [3]) **will translate to significant gains in FID** for large enough models.

### 3. References used in the rebuttal:

[1] Taming Transformers for High-Resolution Image Synthesis, Esser et al., 2020

[2] Language Model Beats Diffusion – Tokenizer is Key to Visual Generation, Yu et al., 2023

[3] Finite Scalar Quantization: VQ-VAE Made Simple, Mentzer et al., 2023

[4] Omnidata: A Scalable Pipeline for Making Multi-Task Mid-Level Vision Datasets from 3D Scans, Eftekhar et al., 2021

[5] SDXL: Improving Latent Diffusion Models for High-Resolution Image Synthesis, Podell et al., 2023

[6] Adding Conditional Control to Text-to-Image Diffusion Models, Zhang et al., 2023

[7] Scaling Autoregressive Models for Content-Rich Text-to-Image Generation, Yu et al., 2022

[8] MaskGIT: Masked Generative Image Transformer, Chang et al., 2022

[9] Scalable Pre-training of Large Autoregressive Image Models, El-Nouby et al., 2024

[10] Exploring the Limits of Transfer Learning with a Unified Text-to-Text Transformer, Raffel et al., 2019

[11] A Generalist Agent, Reed et al., 2022

[12] Fuyu-8B: A Multimodal Architecture for AI Agents (Blog), Adept, 2022

[13] GiT: Towards Generalist Vision Transformer through Universal Language Interface, Wang et al., 2024

[14] ImageBind: One Embedding Space To Bind Them All, Girdhar et al., 2023

[15] GLU Variants Improve Transformer, Shazeer, 2020

[16] Muse: Text-To-Image Generation via Masked Generative Transformers, Chang et al., 2023

[17] MAGE: MAsked Generative Encoder to Unify Representation Learning and Image Synthesis, Li et al., 2022

[18] 4M: Massively Multimodal Masked Modeling, Mizrahi et al., 2023

[19] Chameleon: Mixed-Modal Early-Fusion Foundation Models, Chameleon Team, 2024

---

### Decision · Program_Chairs · 2024-09-25

**Decision:**

Accept (poster)

**Comment:**

This submission presented a multimodal framework trained on a wide range of modalities and tasks. The authors propose to use large-scale datasets including images, text, semantic and geometric modalities, feature maps from state-of-the-art models and new modalities such as image metadata or color. Processing at least three times as many tasks and modalities as existing models, this unified single model delivers excellent results and also improves fine-grained, controllable multimodal generation.
Reviewers asked for several clarifications. After the rebuttal, all were satisfied with the answers.
The reviewers and AC are ultimately convinced by this submission, considering that the novelty and interest of the proposed multimodal model and learning framework are clearly evident.
The authors are strongly encouraged to take all comments into account in their final version.